# BPRL: A Behavioral Approach to State Representation in Reinforcement Learning

## Abstract

Deep Reinforcement Learning (DRL) often suffers from poor sample efficiency and limited generalization in complex, high-dimensional environments. A key challenge is designing effective state representations, which typically requires manual, domain-specific feature engineering. We propose Behavioral Programming Reinforcement Learning (BPRL), a framework that automates the construction of compact, semantically rich state representations. BPRL leverages Behavioral Programming (BP)—a scenario-based modeling paradigm—to specify the environment's dynamics. The core contribution is that the very same BP model used to define the environment's logic is also used to *automatically derive* the state representation for the DRL agent. This dual use of the BP model eliminates the need for manual feature design while ensuring that the extracted representations capture both high-level symbolic structure and temporal dependencies. By combining BP's modularity with structured observations, BPRL simplifies environment modeling and enhances agent learning. Experiments across multiple DRL algorithms on MiniGrid benchmarks demonstrate that BPRL substantially improves sample efficiency and asymptotic performance over standard baselines.

## 1 Introduction

Deep Reinforcement Learning (DRL) is a powerful framework for solving complex sequential decision-making tasks across domains such as robotics, games, healthcare, and autonomous systems. However, when modeling complex, high-dimensional state spaces, it remains sample-inefficient, computationally intensive, often struggling to capture temporal dependencies or high-level structure between state variables, and generalize across tasks (Patel et al., 2023). These limitations hinder its deployment in real-world settings where data is costly and the task structure is complex.

Representation learning in DRL helps address high-dimensional observations by transforming raw inputs into compact, semantically meaningful features. Common methods include end-to-end neural models and embedding networks, with recent work exploring state-space decomposition (Mohan et al., 2024) and singular value decomposition for preserving transition structure (Chandak et al., 2023). Model-based approaches can improve sample efficiency by learning environment dynamics, but often trade off predictive accuracy (Echchahed & Castro, 2025). However, many methods lack support for behavioral or temporal constraints, remain hyperparameter-sensitive, and struggle to incorporate domain knowledge.

Neuro-symbolic RL introduces symbolic reasoning into DRL pipelines to improve interpretability, generalization, and efficiency through logic constraints, symbolic planners, or differentiable inference modules (Acharya et al., 2024). In this study, we introduce a complementary perspective: instead of embedding symbolic reasoning directly into policy learning, we leverage Behavioral Programming (BP)—a well-established scenario-based modeling approach—to model environment dynamics and automatically derive compact, semantically rich state representations.

Our framework, Behavioral Programming Reinforcement Learning (BPRL), integrates BP with DRL, enabling automatic generation of structured observations aligned with system requirements. This approach simplifies environment design and maintenance, while forgoing the complexity of explicit reasoning engines or differentiable logic. Despite its simplicity, it still captures high-level symbolic structure and temporal dependencies, providing advanced multi-modal state representations for RL agents.

We evaluate BPRL on multiple DRL algorithms and diverse environments that vary in complexity. Our experiments demonstrate that BPRL state representations enable both significantly faster convergence and higher final performance, compared to standard baselines. The behavioral decomposition particularly excels in environments with complex temporal dependencies and high-dimensional observation spaces, where traditional methods struggle to identify relevant features efficiently.

**Contributions.** Our contributions are threefold: (1) We propose a novel framework that integrates Behavioral Programming with DRL to enable *automated extraction of structured state representations*, capturing both high-level symbolic structure and temporal dynamics of environments. (2) Our BP-based environment modeling for RL provides modular, incremental, and high-level specification of complex behaviors, thereby simplifying environment design and supporting non-intrusive extensibility. (3) We empirically demonstrate that BPRL significantly improves learning efficiency and performance across multiple RL algorithms and environments, addressing key challenges in scaling RL to complex, high-dimensional tasks.

## 2 PRELIMINARIES

### 2.1 BEHAVIORAL PROGRAMMING (BP)

BP is a well-established paradigm for constructing reactive systems through the incremental composition of desired behaviors (Harel et al., 2011b; Elyasaf, 2021), with robust mathematical semantics and implementations in languages such as JavaScript and Python. This foundation has supported extensive research in fields such as model checking (Harel et al., 2011a), compositional verification (Harel et al., 2013), and synthesis (Kugler et al., 2011). Recent work has also explored the integration of BP with DRL (see Related Work). In BP, developers implement *behavioral threads* (b-threads), each specifying what the system may, must, or must not do based on discrete requirements. For example, a b-thread might enforce that "at each timestep, the agent may move forward or rotate 90° left/right." The collection of such b-threads forms a *behavioral program* (b-program).

Each b-thread can be viewed as a labeled transition system, with each state being a *synchronization point*, at which the b-thread specifies its stance toward upcoming system events by declaring:

- *Requested events:* events the b-thread actively proposes,
- *Blocked events:* events it seeks to prevent,
- *Waited-for events:* events it observes but does not influence.

Once all b-threads reach a synchronization point and place their declarations, they suspend. An event arbitrator selects a single event that was requested and not blocked, and resumes all the b-threads that requested or waited for this event. Upon resuming, they move to their next synchronization point. To illustrate these concepts, we implement several MiniGrid environments using BP.

A concise summary of the BP formalism appears in Appendix H, following the original definitions of Harel et al. (2010). Here, we focus only on the components relevant to our RL framework.

### 2.2 MINIGRID

*MiniGrid* is a suite of lightweight, configurable, grid-world environments widely used in reinforcement learning (RL) research (Chevalier-Boisvert et al., 2023). Each environment presents goal-driven tasks in a discrete space where an agent interacts with objects such as doors, keys, and boxes. The layout is randomized each episode, encouraging generalization rather than memorization. Agents operate under a limited action space, which includes moving forward, rotating, picking up and dropping objects, and opening or closing doors. Representative environments are depicted in Figure 1.

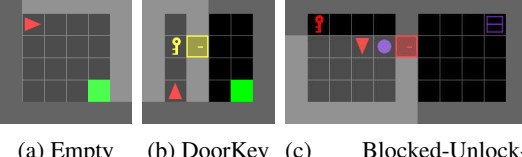

(a) Empty    (b) DoorKey    (c) Blocked-Unlock-Pickup

Figure 1: MiniGrid environments. `Empty` requires the agent to reach a goal. `DoorKey` requires finding a key and opening a door to reach the goal. `Blocked-Unlock-Pickup` requires the agent to move a blocking object, pick up a key, open a door, and pick up a box.

A major challenge in MiniGrid is its sparse reward structure, wherein agents receive a reward only upon task completion:

$$R = \begin{cases} 1 - 0.9 \, \dfrac{\text{step\_count}}{\text{max\_steps}} \,, & \text{if the goal is reached} \\ 0 \,, & \text{otherwise} \end{cases}$$

This leads to difficult long-horizon credit assignment, exploration challenges, and an absence of intermediate feedback signals — all recognized as challenging problems in RL.

## 3 THE PROPOSED METHOD: **BPRL**

This section is organized as follows. We first describe how to implement environments using our BP-based approach, comparing the required manual efforts with those of standard methods. We then elaborate on the BPRL framework and its novel automated state encoding process. Finally, we describe how BPRL can be applied to improve learning in existing non-BP environments.

### 3.1 IMPLEMENTING MINIGRID WITH BP

As is standard in DRL, manual implementation of the environment is a prerequisite. Our method is no different in this regard; we simply propose that this implementation be done using Behavioral Programming (BP) instead of a general-purpose language like Python. Although this approach requires some familiarity with the BP paradigm, we note that recent advancements can lower this barrier, as Large Language Models (LLMs) have proven effective in translating natural language requirements into BP code (Yaacov et al., 2024; Harel et al., 2024).

A key advantage of our approach is *the automation of a subsequent, typically manual, step*. In a standard DRL framework, after defining the environment's logic, the programmer must then manually design and implement the state representation that is fed to the DRL agent (e.g., a vector or a matrix). *Our framework eliminates this manual encoding step entirely* by automatically translating the state of the implemented BP model into a consistent state representation. This automation relieves the user of this engineering effort.

To demonstrate the implementation process, we use the MiniGrid-Empty and MiniGrid-DoorKey environments. These examples show how BP's features facilitate modular and incremental development while also serving to clarify the paradigm's core concepts.

Our implementation is written in BPpy, a Python-based framework for Behavioral Programming (Yaacov, 2020). In this framework, b-threads are implemented as generator functions, and synchronization points are represented by `yield sync(...)` calls, which halt the b-thread until the next event selection. Each sync statement encapsulates the blocked, requested, and waited-for events. The full implementations can be found at `https://anonymous.4open.science/r/BPRL`.

**MiniGrid-Empty.** Listing 1 shows the b-program for the *MiniGrid-Empty* environment. Each b-thread in the program specifies and enforces a particular aspect of the environment's logic—such as blocking agent movement into walls (`wall`), placing the agent at the initial location to start the simulation (`start`), or waiting for the agent to reach the goal before ending the episode (`goal`). Rotation b-threads manage orientation, as detailed in our repository. Notably, the `Move_forward_only` b-thread constrains the agent to move only forward relative to its current orientation, blocking all other movement events by continuously monitoring the agent's state and updating valid actions accordingly.

**MiniGrid-DoorKey.** The MiniGrid-Empty environment features simple behaviors, with greater complexity arising in each subsequent environment. Thanks to BP's modular and incremental nature, we extend the MiniGrid-Empty implementation (Listing 1) to model the more complex *MiniGrid-DoorKey* environment by adding new b-threads (Listing 2). For example, the door's two states (close or open) are handled not by a boolean variable, but through the `door_alternate_open_close` b-thread, which behaviorally alternates between waiting for the door to open (while blocking closing) and waiting for it to close (while blocking opening), starting

from the closed state. This illustrates how BP encodes behavioral logic directly through b-threads, which, as we will demonstrate later, enhances learning.

Similarly, the `door_unlock_with_key` b-thread tracks the agent's interactions with the key: it enables unlocking the door only when the agent possesses the key and reverts to the locked state if the key is dropped before unlocking; once unlocked, the agent may freely open and close the door, regardless of the key. For brevity, some auxiliary behaviors related to the door and key are omitted.

```python
@b_thread
def wall(x, y):
    yield sync(block=Move(x, y))

@b_thread
def start(agent_x, agent_y):
    # Place agent at initial location
    yield sync(request=Move(agent_x, agent_y))

@b_thread
def goal(x,y):
    yield sync(waitFor=Move(x,y))
    yield sync(block=All(), terminated=True)

# .... Rotation b-threads that manage the orientation of the ↩
      agent.

@b_thread
def move_forward_only(initial_orientation):
    orientation = initial_orientation
    event = yield sync(waitFor=any_move_event) # initial ↩
        move event
    x,y = event.data["x"], event.data["y"]

    while True:
        x_f,y_f = get_forward_location(x, y, orientation)
        # All Moves except for forward
        non_forward_moves = Any("Move").not(Move(x_f,y_f))
        event = yield sync(block=non_forward_moves,
            waitFor=[any_move_event, any_rotate_event])
        # .... Update location and orientation based on event
```

Listing 1: A BPpy implementation of the MiniGrid Empty environment. Each b-thread is responsible for a different behavioral aspect, requesting, blocking, and waiting for events at synchronization points. A domain-agnostic execution engine runs these b-threads, creating a cohesive behavior consistent with all of them.

```python
# All b-threads of the empty environment ↩
        (Listing 1) plus the following b-threads:

@b_thread
def door(x,y):
    while True:
        yield sync(block=Move(x,y), waitFor=Open(x,y))
        yield sync(waitFor=Close(x,y))

@b_thread
def door_alternate_open_close(x,y):
    while True:
        yield sync(block=Close(x,y), waitFor=Open(x,y))
        yield sync(block=Open(x,y), waitFor=Close(x,y))

@b_thread
def door_unlock_with_key(x,y):
    while True:
        yield sync(block=Open(x,y),
            waitFor=any_pickup_event)
        e = yield sync(waitFor=[any_drop_event, Open(x,y)])
        if e == Open(x,y):
            return # Door has been unlocked; no further ↩
            intervention is needed
# ...
```

Listing 2: MiniGrid DoorKey environment: This BPpy implementation extends the *MiniGrid-Empty* environment with door-specific behavior. The `door` b-thread blocks movement into the door's cell until it's opened. The `door_alternate_open_close` b-thread enables toggling the door state, and the `door_unlock_with_key` b-thread allows the agent to unlock the door only if it holds a key; it terminates after unlocking, allowing unrestricted door access thereafter. Key-related b-threads are omitted for brevity.

## 3.2 AUTOMATIC STATE REPRESENTATION

A key contribution of our framework is its ability to automatically derive a semantically rich state representation directly from the BP environment model, eliminating the need for manual feature engineering. This process is rooted in the formal semantics of BP.

As we recall, each b-thread is formally a labeled transition system (LTS), where each state corresponds to a synchronization point and each transition is an event. For instance, the `door_alternate_open_close` b-thread can be viewed as a simple LTS with two states representing the door being open or closed, and the Open and Close events trigger transitions between them. Our framework leverages this inherent structure to construct a state vector for the DRL agent.

The encoding process is as follows (illustrated in Figure 2). At any time step $t$, the state of a single b-thread is captured as a one-hot vector, $h^t \in \{0, 1\}^{|S|}$, which indicates its current synchronization

point out of its $|S|$ possible points. To form a complete representation of the behavioral program, we concatenate the vectors from all $n$ b-threads: $o_{\text{BP}}^t = \bigoplus_{i=1}^{n} h_i^t$.

The dimensionality of this concatenated vector is $\sum_i |S_i|$. For practical implementation, we pad each one-hot vector to a uniform maximum length $K$, resulting in a fixed-size state representation of $n \times K$. This simplification preserves modularity and interpretability without significant overhead, as most b-threads are concise, comprising only a few states. We refer to this final vector, $o_{\text{BP}}^t$, as *BP-observation*.

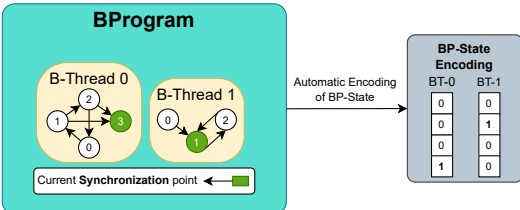

Figure 2: The automated state encoding process. The environment is modeled as a b-program composed of b-threads, where each b-thread is an LTS. The b-program's state is automatically encoded by combining one-hot vectors representing the current state of each b-thread.

This resulting BP-observation is not merely a structural artifact; it is a compact and semantically rich representation. Because a synchronization point defines a b-thread's current status in its event-driven sequence, *the vector captures two critical dimensions of the system's state*: **(i) Temporal Structure:** The progression and ordering of events for a specific behavior. **(ii) Behavioral Intent:** The functional goal or role encoded within each active b-thread.

A central advantage of our approach is that a symbolic, semantically rich representation of behavior is generated without additional programming effort. In standard DRL workflows, one must not only implement the environment's logic but also, as a separate manual step, construct a symbolic abstraction or feature layer if a higher-level representation is desired. The BPRL framework eliminates this second step entirely: *the very same BP model that specifies the environment's dynamics automatically yields the state representation*.

Furthermore, the inherent modularity and incrementality of BP are preserved; new behaviors can be added via new b-threads without refactoring existing code (as shown in listings 1 and 2), and the state representation expands automatically. This combination of automatic representation, low overhead, and scalable design makes BPRL a practical and powerful tool.

### 3.3 Boosting Existing Implementations with BP-Strategies

Beyond implementing an entire environment in BP, our framework also offers a lightweight method to enhance existing, non-BP implementations (e.g., those written in Python). This is achieved by augmenting the environment's native observations with small, targeted b-threads that inject high-level, symbolic knowledge. We refer to these manually-defined b-threads as BP-strategies.

It is crucial to distinguish this approach from the one previously described. In this context, BP is used for *manual feature engineering*, allowing a domain expert to handcraft features based on their knowledge. This contrasts sharply with our core proposal, where *a complete and semantically rich state representation is derived fully and automatically* from the environment's BP specification. We explore this "manual boosting" approach for two key reasons:

1. To showcase BP's flexibility as a powerful tool for incorporating domain-specific knowledge into any DRL agent's observations.

2. To establish a clear baseline for comparing the effectiveness of hand-crafted features against our framework's automated representation.

Our selection of specific strategies was guided by the principle of comparable human effort. The main advantage of using BP for state representation (BP-observation) is the natural alignment of the code to the requirements; the b-threads are essentially a formal representation of the requirement text. To create a fair comparison, we sought a baseline that reflects a similar workflow: just as the BP-observation user translates *requirements* into BP, the BP-strategy user translates *simple heuristics* into BP. While it is possible to engineer sophisticated, highly optimized heuristics to achieve superior results, doing so would misrepresent the comparison. We explicitly sought strategies that represent the natural behavioral cues a user would intuitively reach for—simple heuristics rather than complex algorithmic developments.

To demonstrate this flexibility, we implemented four BP-strategies. Two are simple heuristics reflecting expert knowledge about efficient navigation: (i) counting consecutive clockwise turns and (ii) counting consecutive counterclockwise turns. These features help the agent recognize redundant patterns such as repeatedly turning in the same direction or immediately undoing a movement.

The other two are more complex, task-specific features: (iii) tracking the agent's distance to its current objective (key or door), and (iv) encoding the current high-level task phase (e.g., "search for key" phase followed by "search for door" phase). Together, these provide explicit, high-level context about progress toward the goal.

All strategies and their observation vectors were manually designed and concatenated with the original MiniGrid observation along the spatial dimension, yielding an extended representation that integrates raw sensory input with high-level procedural cues.

## 4 EXPERIMENTS

We evaluate `BPRL` by integrating it into two popular DRL algorithms: Proximal Policy Optimization (PPO) (Schulman et al., 2017) and Advantage Actor-Critic (A2C) (Mnih et al., 2016). We evaluated the algorithms' performance in multiple MiniGrid environments. All experiments were carried out using the Stable-Baselines3 (SB3) library (Raffin et al., 2021).

### 4.1 BASELINES

The goal of our evaluation is to determine whether our BP-based state representation is effective. To this end, we augment the state representation used by PPO and A2C with several variants of our approach, while also comparing our variants to existing state augmentation methods:

- *Original observation:* The standard MiniGrid observation, consisting of a $H \times W \times 3$ fully observable grid encoding cell contents (object type, color, and state).
- *Frame-stacked original observation:* A sequence of four recent original observations stacked along the channel dimensions. Used to provide temporal context.
- *Recurrent PPO with original observation:* A variant of PPO that incorporates a recurrent network to model temporal dependencies across observation sequences. This agent was trained using the standard MiniGrid observation and was not used with A2C.

**BP-based Observation Types.** The standard PPO and A2C algorithms received the following variants of our BP-based state representations:

- *Original + strategies:* Original observation augmented with the manually defined BP-strategies (manual feature construction using BP).
- *BP-derived observation (BP-observation):* An *automatically generated* state representation based solely on the internal state of the behavioral program (without the original observation).
- *BP-observation + original:* BP-observation concatenated with Original observation.
- *BP-observation + original + BP-strategies:* A combined representation concatenating the original observation, BP-observation and BP-strategies (automatic AND manual).

### 4.2 EXPERIMENTAL SETUP & METRICS

For the DRL agents, we used a fixed neural network architecture: a CNN for feature extraction followed by a fully connected network (FCN) for policy and value prediction. To ensure a fair comparison, all experiments were conducted with the same hyperparameters across environments. We evaluated our approach with six MiniGrid environments of varying complexity: `Empty 6x6`, `DoorKey 6x6`, `DoorKey 8x8`, `Unlock`, `Unlock-Pickup`, and `Blocked-Unlock-Pickup`. Additional details on these environments are provided in Appendix A of the supplementary materials.

BP-observations were first processed through a dedicated two-layer FCN. When used alone, the resulting feature vector was passed directly to the main FCN for policy and value prediction. When combined with standard MiniGrid observations, these were first encoded by a CNN, and the resulting feature vector was concatenated with the BP-observation FCN output before being passed to

| PPO agent | Empty 6x6 | Unlock | DoorKey 6x6 | DoorKey 8x8 | Unlock-Pickup | Blocked-Unlock-Pickup |
|---|---|---|---|---|---|---|
| Original | 0.97 ± 0.007 | 0.971 ± 0.015 | 0.938 ± 0.063 | 0.958 ± 0.01 | 0.849 ± 0.249 | 0.001 ± 0.001 |
| Frame-Stack | 0.981 ± 0.014 | 0.973 ± 0.002 | **0.968 ± 0.002** | 0.942 ± 0.071 | 0.06 ± 0.085 | 0.0 ± 0.001 |
| RPPO with Original | 0.948 ± 0.006 | 0.843 ± 0.148 | 0.932 ± 0.172 | 0.733 ± 0.322 | 0.067 ± 0.149 | 0.0 ± 0.001 |
| Original + Strategies | 0.969 ± 0.005 | 0.963 ± 0.025 | 0.943 ± 0.061 | 0.951 ± 0.015 | 0.951 ± 0.015 | 0.017 ± 0.083 |
| BP-observation | 0.983 ± 0.022 | 0.974 ± 0.001$^{\ddagger}$ | 0.963 ± 0.005 | 0.887 ± 0.185 | 0.955 ± 0.003 | 0.396 ± 0.456$^{\dagger\ddagger}$ |
| BP-observation + Original | **0.987 ± 0.006** | **0.975 ± 0.015** | 0.964 ± 0.018 | **0.965 ± 0.005$^{\ddagger*}$** | 0.951 ± 0.055$^{\dagger\ddagger}$ | 0.408 ± 0.459$^{\dagger\ddagger}$ |
| BP-obs + Original + Strategies | 0.984 ± 0.011$^{\ddagger*}$ | 0.944 ± 0.141 | 0.966 ± 0.012$^{\dagger}$ | 0.957 ± 0.013$^{\ddagger}$ | **0.959 ± 0.011$^{\dagger\ddagger}$** | **0.485 ± 0.454$^{\dagger\ddagger}$** |

| A2C agent | Empty 6x6 | Unlock | DoorKey 6x6 | DoorKey 8x8 | Unlock-Pickup |
|---|---|---|---|---|---|
| Original | 0.989 ± 0.002$^{\ddagger}$ | 0.055 ± 0.021 | 0.074 ± 0.038 | 0.012 ± 0.01 | 0.003 ± 0.002 |
| Frame-Stack | 0.81 ± 0.017 | 0.052 ± 0.015 | 0.05 ± 0.009 | 0.033 ± 0.035 | 0.003 ± 0.002 |
| Original + Strategies | 0.862 ± 0.011 | 0.051 ± 0.009 | 0.109 ± 0.056 | 0.127 ± 0.183$^{\dagger}$ | 0.005 ± 0.004 |
| BP-observation | 0.991 ± 0.002$^{\dagger\ddagger}$ | 0.946 ± 0.097$^{\dagger\ddagger}$ | 0.877 ± 0.257$^{\dagger\ddagger}$ | 0.907 ± 0.017$^{\dagger\ddagger}$ | 0.848 ± 0.215$^{\dagger\ddagger}$ |
| BP-observation + Original | **0.992 ± 0.0$^{\dagger\ddagger}$** | **0.974 ± 0.006$^{\dagger\ddagger}$** | 0.856 ± 0.229$^{\dagger\ddagger}$ | **0.954 ± 0.02$^{\dagger\ddagger}$** | **0.938 ± 0.025$^{\dagger\ddagger}$** |
| BP-obs + Original + Strategies | **0.992 ± 0.0$^{\dagger\ddagger}$** | **0.974 ± 0.006$^{\dagger\ddagger}$** | **0.975 ± 0.005$^{\dagger\ddagger}$** | 0.935 ± 0.025$^{\dagger\ddagger}$ | 0.937 ± 0.032$^{\dagger\ddagger}$ |

"Significantly better" denotes statistical significance according to paired t-tests ($p < 0.05$).

[*] Significantly better than *Original* observation.

[†] Significantly better than all non-BP baselines (*Original, Frame-Stack, RPPO*).

[‡] Significantly better than *Original + Strategies*.

Table 1: PPO and A2C performance: Mean ± Std final reward across environments for each configuration. Best-performing configurations per environment are bolded.

the main FCN. The neural architecture and hyperparameters were optimized on the original Mini-Grid observations, and the same configuration was then applied in our BPRL experiments to ensure comparability. Full architectural details are provided in the supplementary material (sections B, C).

We evaluated performance by comparing *sample efficiency*, *convergence rate*, and *maximum reward* across the different configurations. For each configuration, metrics were computed as the mean and standard deviation of episode rewards, averaged over 30 or more runs with different random seeds.

## 4.3 RESULTS

Our results for the PPO and A2C algorithms are presented in Table 1. Across almost all environments, BP-based state representations outperform baseline configurations, with agents consistently achieving higher final rewards. Notably, the manually crafted BP-strategies (*Original + Strategies*) underperform relative to automatically generated BP-derived observations. For A2C, we evaluated all environments, but none of the configurations succeeded in solving the `Blocked-Unlock-Pickup` task, and therefore it is omitted from the table. Paired t-tests confirm that BP-derived observations significantly outperform baselines in most environments ($p < 0.05$), and this statistical evidence is complemented by a substantial practical advantage: agents using BP-observations demonstrate markedly improved *sample efficiency*, reaching high performance with far fewer interactions than the baselines. This efficiency gain, analyzed in more detail in the following paragraphs, underscores the strength and high applicability of BPRL to a range of DRL problems.

Our experimental results support our central claim: encoding environment logic via BP-state representations yields compact, task-relevant features that enable more sample-efficient and stable learning in sequential decision-making tasks.

## 4.4 ANALYZING CONVERGENCE SPEED

Our BP-based observations enable faster learning and convergence in DRL algorithms. As shown in Table 2, BP-based methods—especially *BP-observations + Original* and *BP-obs + Original + Strategies*—consistently achieve significantly higher rewards than non-BP baselines throughout training steps, for both PPO and A2C. For example, PPO's BP-observation variant reaches a mean reward of 0.409 after 150K steps, compared to 0.122 for the best non-BP baseline (a 230% increase), while A2C shows over a 600% improvement on the same task at 600K steps. These large margins highlight the improved *sample efficiency* of BP-derived observations: agents require substantially fewer interactions with the environment to achieve strong performance. Figure 3 visually reinforces this advantage, showing that BP-based methods converge faster and more consistently across tasks,

| **PPO agent** | Unlock | | | | | Unlock Pickup | | | | |
|---|---|---|---|---|---|---|---|---|---|---|
| | R@50k | R@150k | R@450k | R@700k | % Solved | R@500k | R@1 m | R@2.5 m | R@3.75 m | % Solved |
| Original | $0.055 \pm 0.018$ | $0.122 \pm 0.050$ | $0.579 \pm 0.103$ | $0.958 \pm 0.015$ | 100.000 | $0.003 \pm 0.003$ | $0.002 \pm 0.003$ | $0.216 \pm 0.298$ | $0.807 \pm 0.280$ | 86.538 |
| Frame-Stack | $0.048 \pm 0.015$ | $0.089 \pm 0.042$ | $0.761 \pm 0.095$ | $0.954 \pm 0.009$ | 100.000 | $0.002 \pm 0.003$ | $0.005 \pm 0.004$ | $0.023 \pm 0.034$ | $0.057 \pm 0.080$ | 0.000 |
| RPPO with Original | $0.045 \pm 0.021$ | $0.060 \pm 0.028$ | $0.182 \pm 0.117$ | $0.498 \pm 0.226$ | 74.194 | $0.003 \pm 0.004$ | $0.003 \pm 0.005$ | $0.006 \pm 0.011$ | $0.034 \pm 0.108$ | 0.000 |
| Original + Strategies | $0.050 \pm 0.021$ | $0.231 \pm 0.095$ | $0.908 \pm 0.028$ | $0.964 \pm 0.028$ | 100.000 | $0.038 \pm 0.057$ | $0.267 \pm 0.215$ | $0.911 \pm 0.133$ | $0.948 \pm 0.022$ | 100.000 |
| BP-observation | $0.048 \pm 0.020$ | $\mathbf{0.409 \pm 0.131}$ | $0.969 \pm 0.003$ | $0.974 \pm 0.002$ | 100.000 | $0.241 \pm 0.173$ | $0.785 \pm 0.325$ | $\mathbf{0.954 \pm 0.008}$ | $0.956 \pm 0.004$ | 100.000 |
| BP-observation + Original | $0.050 \pm 0.020$ | $0.327 \pm 0.105$ | $0.951 \pm 0.107$ | $0.967 \pm 0.074$ | 100.000 | $0.216 \pm 0.199$ | $0.745 \pm 0.366$ | $0.914 \pm 0.183$ | $0.948 \pm 0.093$ | 98.000 |
| BP-obs + Original + Strategies | $\mathbf{0.056 \pm 0.024}$ | $0.338 \pm 0.124$ | $\mathbf{0.970 \pm 0.027}$ | $\mathbf{0.976 \pm 0.010}$ | 100.000 | $\mathbf{0.310 \pm 0.193}$ | $\mathbf{0.865 \pm 0.229}$ | $0.937 \pm 0.086$ | $\mathbf{0.958 \pm 0.015}$ | 100.000 |

| **A2C agent** | Unlock | | | | | Unlock Pickup | | | | |
|---|---|---|---|---|---|---|---|---|---|---|
| | R@300k | R@600k | R@1.50 m | R@2.25 m | % Solved | R@1 m | R@2 m | R@5 m | R@7.50 m | % Solved |
| Original | $0.052 \pm 0.015$ | $0.059 \pm 0.019$ | $0.053 \pm 0.019$ | $0.058 \pm 0.016$ | 13.333 | $0.002 \pm 0.001$ | $0.003 \pm 0.001$ | $0.005 \pm 0.002$ | $0.001 \pm 0.001$ | 0.000 |
| Frame-Stack | $0.059 \pm 0.023$ | $0.061 \pm 0.021$ | $0.045 \pm 0.020$ | $0.063 \pm 0.016$ | 6.667 | $0.003 \pm 0.001$ | $0.003 \pm 0.001$ | $0.004 \pm 0.001$ | $0.002 \pm 0.001$ | 0.000 |
| Original + Strategies | $0.052 \pm 0.016$ | $0.046 \pm 0.016$ | $0.053 \pm 0.015$ | $0.058 \pm 0.020$ | 0.000 | $0.005 \pm 0.002$ | $0.003 \pm 0.001$ | $0.004 \pm 0.001$ | $0.005 \pm 0.002$ | 0.000 |
| BP-observation | $0.145 \pm 0.063$ | $0.406 \pm 0.133$ | $\mathbf{0.920 \pm 0.037}$ | $\mathbf{0.971 \pm 0.004}$ | 100.000 | $0.007 \pm 0.003$ | $0.009 \pm 0.004$ | $0.146 \pm 0.059$ | $0.671 \pm 0.112$ | 85.000 |
| BP-observation + Original | $0.139 \pm 0.070$ | $0.438 \pm 0.086$ | $0.913 \pm 0.032$ | $0.966 \pm 0.003$ | 100.000 | $\mathbf{0.009 \pm 0.004}$ | $0.018 \pm 0.006$ | $\mathbf{0.379 \pm 0.084}$ | $0.814 \pm 0.095$ | 100.000 |
| BP-obs + Original + Strategies | $\mathbf{0.178 \pm 0.069}$ | $\mathbf{0.446 \pm 0.095}$ | $0.911 \pm 0.034$ | $0.965 \pm 0.003$ | 100.000 | $0.007 \pm 0.003$ | $\mathbf{0.025 \pm 0.007}$ | $0.369 \pm 0.079$ | $\mathbf{0.824 \pm 0.098}$ | 100.000 |

Table 2: The mean rewards of each baseline after a fixed number of steps ($R@\{steps\}$) is used to measure the convergence rate of each algorithm. *% Solved* is the fraction of runs in which the agent reached a reward of over 0.9. We present the results for the *Unlock* and *Unlock-Pickup* environments. Results for all of the environments appear in supplementary material, Appendix E.

reaching high rewards in considerably fewer training steps. Taken together, these results underscore the effectiveness of BP-derived observations in improving the efficiency of DRL training.

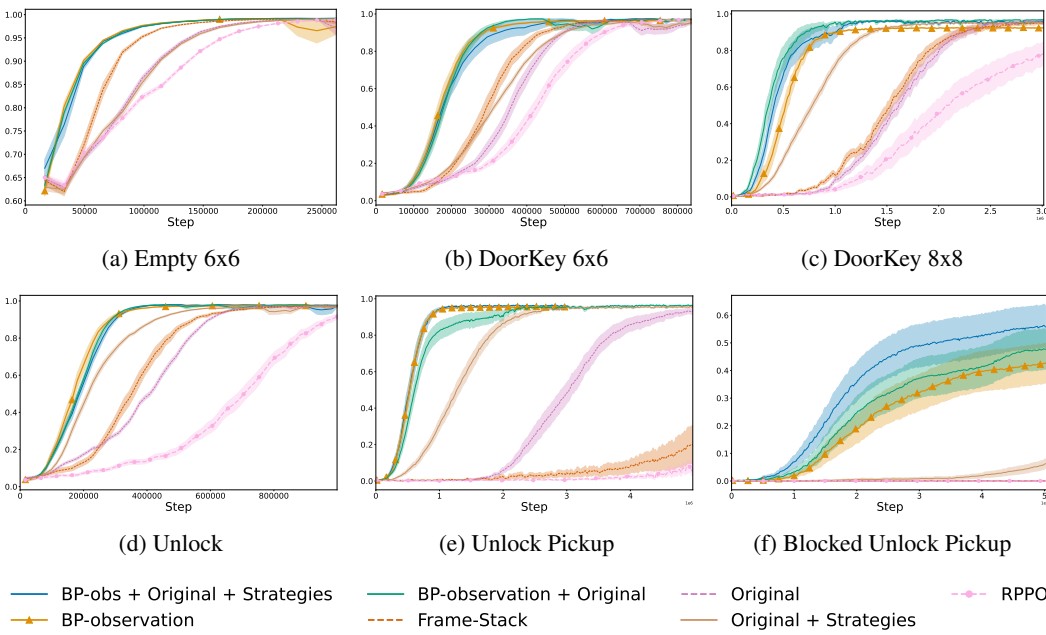

(a) Empty 6x6     (b) DoorKey 6x6     (c) DoorKey 8x8

(d) Unlock     (e) Unlock Pickup     (f) Blocked Unlock Pickup

- BP-obs + Original + Strategies
- BP-observation
- BP-observation + Original
- Frame-Stack
- Original
- Original + Strategies
- RPPO

Figure 3: Mean episode reward and standard error (shaded) of PPO agents throughout training across various MiniGrid environments, comparing different observation configurations and Recurrent PPO. Corresponding results for A2C agents are provided in the appendix.

# 5 RELATED WORK

## 5.1 REPRESENTATION LEARNING FOR STATE AND ACTION SPACES

Multiple studies propose methods for representing and reducing action spaces to improve DRL performance. In Chandak et al. (2019), the authors demonstrate how policies can be decomposed into components acting in low-dimensional action spaces, improving generalization over large, finite action sets. The work of Dulac-Arnold et al. (2015) tackles domains with large numbers of discrete actions by embedding them in continuous spaces upon which agents can generalize, using approximate nearest-neighbor methods to achieve logarithmic-time lookup complexity. Identifying

and removing redundant actions based on causal effects (Liu et al., 2025) has been shown to improve exploration efficiency and learning performance. Jian et al. (2017) propose a framework for embedding categorical variables, which is highly relevant to discrete action spaces in RL.

Efficient representation of states and actions is central to improving generalization, exploration, and learning efficiency in DRL. For state representation, Liu et al. (2022) introduce an embedding network to merge similar states and use state binary code to count state visits, encouraging exploration through auxiliary rewards. Doerr et al. (2018) propose a variational method to train probabilistic models that capture temporal correlations, while Luis et al. (2024) incorporate a Kalman filter layer for uncertainty-aware inference. In model-based RL, PlaNet Hafner et al. (2019) learns dynamics from image observations via latent stochastic and deterministic models, while Ha & Schmidhuber (2018) compress spatial-temporal information to construct internal models for planning.

For joint state-action representations, Pritz et al. (2021) propose an embedding approach that bridges model-based and model-free methods. Fujimoto et al. (2023) introduce SALE, modeling fine-grained state-action interactions to improve performance in continuous control tasks with TD3, and Yan et al. (2024) present STAR to capture relationships in visually rich tasks. Zambaldi et al. (2019) incorporate relational inductive biases into representation learning by learning state-action embeddings that capture relational structures in the environment.

## 5.2 BP-BASED APPROACHES

The integration of BP with RL has been gaining attention as a way to inject modularity, domain knowledge, and constraints into the learning process. The central motivation is to leverage BP's scenario-based modeling to define behavioral specifications that guide or augment learning. For instance, BP has been used to create high-level behavior modules in robotics, ensuring DRL agents learn policies aligned with complex behavioral patterns (Elyasaf et al., 2019). BP has also been employed to inject expert knowledge into an agent's reward structure, improving adaptability to new requirements without retraining (Yerushalmi et al., 2022). Scenario-based programming was introduced for constrained RL in robotics, enabling expert-defined safety and efficiency constraints and yielding agents that were both high-performing and compliant (Corsi et al., 2022). Beyond specific constraints, BP has also been proposed as a unifying abstraction to integrate formal methods with DRL, guiding policy learning to avoid undesired behaviors (Yaacov et al., 2025).

Unlike prior approaches that use BP to shape rewards or constrain behavior, we use BP to model the environment, leveraging its procedural structure to automatically extract symbolic, task-relevant state representations. This shifts BP's role from policy supervision to observation construction, enabling agents to learn from states that reflect the environment's modular and procedural logic.

## 5.3 FOUNDATIONAL ALGORITHMS AND STRUCTURED REPRESENTATIONS

Several foundational algorithms have shaped the landscape of representation learning in DRL. Lillicrap et al. (2015) introduce DDPG, adapts Q-learning to continuous action spaces using a model-free, off-policy actor-critic approach. SAC (Haarnoja et al., 2018) achieves sample-efficient and robust performance by learning stochastic policies in continuous action spaces. Other approaches focus on the state representation itself, such as learning structured world models that explicitly capture objects and their interactions (Kipf et al., 2019). Separately, HER (Andrychowicz et al., 2017) provides a novel way to learn from failures by relabeling them as successful attempts toward different goals, thus structuring the learning process more effectively.

## 5.4 SYMBOLIC TASK SPECIFICATIONS AND STATE ABSTRACTIONS IN RL

Neuro-symbolic RL combines symbolic reasoning with neural policies to improve interpretability, generalization, and sample efficiency (Acharya et al., 2024). Many such methods integrate symbolic rules, temporal logic constraints, or planning structures into learning—typically through reward shaping, automaton conditioning, or action filtering.

A major line of work specifies tasks using temporal logics and automata. Reward Machines (Icarte et al., 2018) represent non-Markovian rewards as finite-state machines, enabling structured exploration and decomposition. Similarly, Camacho et al. (2019) compile LTL and related languages into

automata that shape the reward function, while Hasanbeig et al. (2021) synthesize automata from experience to segment long-horizon behaviors. These approaches rely on explicit logical specifications or synthesized automata that subsequently guide reward modeling or exploration, rather than constructing the agent's observation space.

Another direction focuses on symbolic or neuro-symbolic state abstraction. Bai et al. (2016) derive Markovian state abstractions via hierarchical planning, and Abductive Abstract RL (Wang et al., 2025) induces high-level sub-MDP abstractions from raw input via abductive reasoning. Recent work also embeds task automata directly into neural models to represent symbolic goals (Yalcinkaya et al., 2025). These methods depend on symbolic predicates, logical inference, or learned a finite-state structure to build abstract state spaces.

In contrast, BPRL employs the Behavioral Programming (BP) model, which defines the environment itself to generate the agent's state representation. Rather than inferring or specifying symbolic abstractions, BPRL directly exposes the internal synchronization state of each behavioral thread as part of the observation. This provides a structured, modular, and semantically meaningful representation without requiring formal task specifications, automaton synthesis, or predicate learning.

BPRL thus differs from prior neuro-symbolic approaches by deriving state features from the operational structure of the environment's implementation, rather than from reward design or high-level logical descriptions.

## 6 CONCLUSIONS AND FUTURE WORK

In this work, we introduced **BPRL**, a framework that bridges high-level environment specification with the state representation needs of Deep Reinforcement Learning. By leveraging Behavioral Programming (BP), **BPRL** infuses DRL agents with procedural and symbolic knowledge through two main applications: (1) the automatic derivation of a structured state representation from a BP model, and (2) the use of targeted BP-strategies for manual feature engineering. Beyond representation, BP also simplifies environment design itself, allowing tasks to be specified incrementally and modularly. Our results confirm that using BP facilitates modular design and yields a task-relevant state representation that enhances learning efficiency, particularly in long-horizon tasks.

In future work, we plan to extend **BPRL** to continuous and partially observable domains.

## 7 REPRODUCIBILITY AND LLM USAGE

To ensure reproducibility, our source code is available anonymously (`https://anonymous.4open.science/r/BPRL`), with all hyperparameters and model architectures detailed in the appendix. LLM was used for copy-editing to improve grammar and phrasing, but not for generating scientific content or code.

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

# Supplementary Material

## A    MINIGRID ENVIRONMENTS

MiniGrid (Chevalier-Boisvert et al., 2023) is a widely used benchmark suite for evaluating reinforcement learning agents in partially or fully observable grid-worlds. Environments are discrete and lightweight, yet designed to capture key challenges such as exploration, delayed rewards, and compositional reasoning. Tasks typically require an agent to navigate, manipulate objects, or satisfy sequential dependencies (e.g., obtaining a key before opening a door). Observations are commonly image-based grid representations. Rewards are sparse, typically granted only upon task completion, encouraging agents to learn efficient exploration and planning strategies.

**Agent setting.**   The agent operates in a discrete action space consisting of `move forward`, `rotate left`, `rotate right`, `pick up`, `drop`, and `toggle` (used for opening and closing doors). Object manipulation actions are constrained by orientation: an object may only be picked up if the agent is facing it directly.

**Environments used.**   In our experiments, we used six environments of increasing difficulty:

- **Empty 6x6.** A baseline navigation task in which the agent must reach a goal placed randomly in a $6 \times 6$ grid. This environment provides a simple setting to validate learning performance under minimal complexity.

- **Unlock.** The agent must locate and pick up a key and use it to unlock a door, after which the episode terminates.

- **DoorKey 6x6.** The agent must locate a key and use it to open a door before reaching the goal.

- **DoorKey 8x8.** A larger variant of the DoorKey task, increasing the exploration burden due to a bigger state space while preserving the same temporal dependency structure.

- **Unlock-Pickup.** A more complex extension of Unlock, where the agent must first find and pick up a key, unlock a door, and then pick up a box located beyond the door. This requires handling multiple sequential dependencies within one episode.

- **Blocked-Unlock-Pickup.** The most challenging task in our study. The agent must first pick up and move an obstructing object that blocks access to the key, then obtain the key, unlock a door, and finally pick up a box. This environment combines navigation, manipulation, and sequential constraints, making it particularly demanding for DRL agents.

These environments were chosen to provide a progression from simple navigation (Empty 6x6) to increasingly complex, temporally structured tasks (Unlock-Pickup and Blocked-Unlock-Pickup). Full implementation details, including initial conditions and object placements, are provided in the supplementary material.

## B    ARCHITECTURE

The DRL agent architecture comprises a feature extractor followed by a Fully Connected Network (FCN) for policy and value prediction. The feature extractor adopts a dual-branch design to encode both visual and procedural information, which is described in this section.

- **Visual branch (CNN):**
    - Input: single image or stack of frames
    - 4 convolutional layers with output channels: 64, 128, 256, 512
    - Each layer: kernel size 3, padding 1, followed by ReLU
    - MaxPooling layer: kernel size 2
    - Output is flattened and projected to a 512-dimensional vector via a fully connected layer with ReLU

- **BP-observation branch (MLP):**
  - Input: one-hot encoded b-threads state vector
  - Flattened and passed through two linear layers:
    * Linear(state-vector-size, 256) $\rightarrow$ ReLU
    * Linear(256, 128) $\rightarrow$ ReLU
  - Output is a 128-dimensional *BP feature vector*
- **Fusion:**
  - When both branches are present, the CNN and BP feature vector are concatenated
  - The combined representation is used as input to the policy and value networks
  - If the visual input is absent, the *BP feature vector* is passed through an additional linear layer before being passed as input to the policy and value networks:
    * Linear(128, 512) $\rightarrow$ ReLU

## C  HYPERPARAMETERS

We use agent implementations from Stable-Baselines3 version 2.0.0. Each experiment was initialized with a random seed using the library's default seeding mechanism. Unless otherwise noted, all hyperparameters not explicitly listed below use the default values provided by the respective implementation. All experiments were conducted using 16 parallel environments.

### C.1 – PPO

For the PPO agent (including RPPO), we found the following hyperparameters, after thorough tuning, to perform well across all environments:

- Learning rate: $1 \times 10^{-4}$
- Discount factor $\gamma$: 0.99
- Clip range: 0.2
- GAE parameter $\lambda$: 0.98
- Optimizer epochs per update: 10
- Gradient clipping: 1
- Number of steps per environment per update (`n_steps`): 1024
- Policy and value network architecture: (256, ReLU, 256, ReLU)
- Batch size: 512

### C.2 – A2C

For the A2C agent, the following hyperparameters were selected based on empirical performance. Separate configurations were determined through environment-specific tuning experiments.

**For *Empty*, *Unlock*, and *DoorKey 6x6*:**

- Learning rate: $5 \times 10^{-4}$
- Entropy coefficient: 0.05
- Discount factor $\gamma$: 0.99
- Number of steps per environment per update (`n_steps`): 128
- Policy and value network architecture: (256, ReLU, 256, ReLU)
- Gradient clipping: 1
- Normalize advantage: True

**For *Unlock-Pickup* and *DoorKey 8x8*:**

- Learning rate: $3 \times 10^{-4}$

- Entropy coefficient: 0.005

- Discount factor $\gamma$: 0.99

- Number of steps per environment per update (`n_steps`): 128

- Policy and value network architecture: (256, ReLU, 256, ReLU)

- Gradient clipping: 1

- Normalize advantage: True

## C.3 NUMBER OF STEPS

The total number of training steps per environment was selected based on the complexity of the task and the performance of the agents. Simpler environments required fewer steps to converge, whereas more challenging tasks necessitated extended training. Table 3 summarizes the approximate number of global environment steps used for each environment-agent pair. Episodes in all environments were subject to a maximum length of 500 steps, resulting in truncation once this limit was reached.

| Environment | PPO | A2C |
|---|---|---|
| Empty 6x6 | 250,000 | 1,000,000 |
| Unlock | 1,000,000 | 3,000,000 |
| DoorKey 6x6 | 1,000,000 | 2,750,000 |
| DoorKey 8x8 | 3,000,000 | 6,500,000 |
| Unlock-Pickup | 5,000,000 | 10,000,000 |
| Blocked-Unlock-Pickup | 5,000,000 | N/A |

Table 3: Number of training steps per environment for PPO and A2C agents.

# D  A2C LEARNING CURVES

**Supplementary A2C Results**. Due to space constraints, the full A2C training curves are presented here (Figure 4). These results align with the trends observed in the PPO experiments, demonstrating smoother learning dynamics and improved final performance when the observation includes the automatically derived BP-observation.

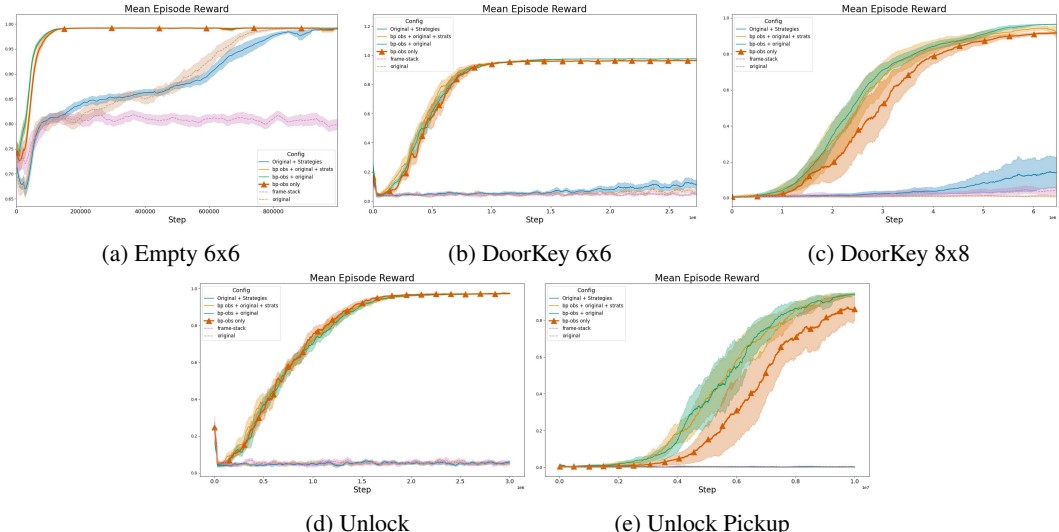

(a) Empty 6x6        (b) DoorKey 6x6        (c) DoorKey 8x8

(d) Unlock        (e) Unlock Pickup

Figure 4: Mean episode reward and standard error (shaded) of A2C agents throughout training across various MiniGrid environments, comparing the different observation configurations. Each curve averages approximately 30 runs.

# E  PPO AND A2C DETAILED RESULTS

This section reports the performance of PPO and A2C agents across all configurations and environments. The tables present the mean episodic reward after specified training steps. We denote this as *R@steps*, where the value corresponds to the mean episodic reward at the indicated number of steps. In addition, *% Solved* indicates the proportion of runs that achieved a final reward above the threshold of 0.9.

## E.1  –PPO

| Configuration | R@50k | R@100k | R@150k | % Solved |
|---|---|---|---|---|
| Original | 0.686 ± 0.030 | 0.862 ± 0.022 | 0.963 ± 0.007 | 100.000 |
| Frame-Stack | 0.708 ± 0.041 | 0.953 ± 0.007 | 0.983 ± 0.001 | 100.000 |
| RPPO with Original | 0.697 ± 0.029 | 0.820 ± 0.021 | 0.920 ± 0.017 | 100.000 |
| Original + Strategies | 0.694 ± 0.033 | 0.852 ± 0.028 | 0.958 ± 0.009 | 100.000 |
| BP-observation | **0.893 ± 0.023** | **0.976 ± 0.002** | **0.988 ± 0.001** | 100.000 |
| BP-observation + Original | 0.872 ± 0.063 | 0.962 ± 0.034 | 0.985 ± 0.007 | 100.000 |
| BP-obs + Original + Strategies | 0.841 ± 0.101 | 0.946 ± 0.056 | 0.979 ± 0.016 | 100.000 |

Table 4: PPO performance on the Empty 6x6 environment

| Configuration | R@50k | R@150k | R@450k | R@700k | % Solved |
|---|---|---|---|---|---|
| Original | 0.055 ± 0.018 | 0.122 ± 0.050 | 0.579 ± 0.103 | 0.958 ± 0.015 | 100.000 |
| Frame-Stack | 0.048 ± 0.015 | 0.089 ± 0.042 | 0.761 ± 0.095 | 0.954 ± 0.009 | 100.000 |
| RPPO with Original | 0.045 ± 0.021 | 0.060 ± 0.028 | 0.182 ± 0.117 | 0.498 ± 0.226 | 74.194 |
| Original + Strategies | 0.050 ± 0.021 | 0.231 ± 0.095 | 0.908 ± 0.028 | 0.964 ± 0.028 | 100.000 |
| BP-observation | 0.048 ± 0.020 | **0.409 ± 0.131** | 0.969 ± 0.003 | 0.974 ± 0.002 | 100.000 |
| BP-observation + Original | 0.050 ± 0.020 | 0.327 ± 0.105 | 0.951 ± 0.107 | 0.967 ± 0.074 | 100.000 |
| BP-obs + Original + Strategies | **0.056 ± 0.024** | 0.338 ± 0.124 | **0.970 ± 0.027** | **0.976 ± 0.010** | 100.000 |

Table 5: PPO performance on the Unlock environment

| Configuration | R@50k | R@150k | R@400k | R@600k | % Solved |
|---|---|---|---|---|---|
| Original | 0.044 ± 0.016 | 0.082 ± 0.039 | 0.590 ± 0.190 | 0.955 ± 0.019 | 100.000 |
| Frame-Stack | 0.038 ± 0.012 | 0.073 ± 0.031 | 0.840 ± 0.073 | 0.961 ± 0.006 | 100.000 |
| RPPO with Original | 0.044 ± 0.013 | 0.103 ± 0.035 | 0.403 ± 0.204 | 0.864 ± 0.188 | 96.667 |
| Original + Strategies | **0.047 ± 0.019** | 0.110 ± 0.039 | 0.737 ± 0.095 | 0.947 ± 0.019 | 100.000 |
| BP-observation | 0.037 ± 0.019 | **0.339 ± 0.198** | **0.946 ± 0.022** | **0.963 ± 0.003** | 100.000 |
| BP-observation + Original | 0.040 ± 0.018 | 0.267 ± 0.170 | 0.936 ± 0.108 | 0.940 ± 0.079 | 100.000 |
| BP-obs + Original + Strategies | 0.041 ± 0.021 | 0.309 ± 0.184 | 0.904 ± 0.203 | 0.953 ± 0.065 | 100.000 |

Table 6: PPO performance on the DoorKey 6x6 environment

| Configuration | R@300k | R@600k | R@1.50m | R@2.25m | % Solved |
|---|---|---|---|---|---|
| Original | 0.016 ± 0.012 | 0.019 ± 0.023 | 0.421 ± 0.187 | 0.905 ± 0.127 | 100.000 |
| Frame-Stack | 0.007 ± 0.008 | 0.026 ± 0.032 | 0.486 ± 0.210 | 0.885 ± 0.156 | 93.939 |
| RPPO with Original | 0.011 ± 0.012 | 0.011 ± 0.015 | 0.216 ± 0.269 | 0.533 ± 0.388 | 61.290 |
| Original + Strategies | 0.047 ± 0.050 | 0.303 ± 0.197 | 0.922 ± 0.060 | 0.951 ± 0.015 | 100.000 |
| BP-observation | 0.127 ± 0.171 | 0.634 ± 0.226 | 0.885 ± 0.184 | 0.889 ± 0.185 | 96.000 |
| BP-observation + Original | **0.242 ± 0.202** | 0.704 ± 0.348 | 0.938 ± 0.112 | **0.964 ± 0.005** | 100.000 |
| BP-obs + Original + Strategies | 0.170 ± 0.159 | **0.724 ± 0.296** | **0.955 ± 0.024** | 0.955 ± 0.017 | 100.000 |

Table 7: PPO performance on the DoorKey 8x8 environment

| Configuration | R@450k | R@950k | R@2.45m | R@3.70m | % Solved |
|---|---|---|---|---|---|
| Original | 0.003 ± 0.003 | 0.002 ± 0.003 | 0.216 ± 0.298 | 0.807 ± 0.280 | 86.538 |
| Frame-Stack | 0.002 ± 0.003 | 0.005 ± 0.004 | 0.023 ± 0.034 | 0.057 ± 0.080 | 0.000 |
| RPPO with Original | 0.003 ± 0.004 | 0.003 ± 0.005 | 0.006 ± 0.011 | 0.034 ± 0.108 | 0.000 |
| Original + Strategies | 0.038 ± 0.057 | 0.267 ± 0.215 | 0.911 ± 0.133 | 0.948 ± 0.022 | 100.000 |
| BP-observation | 0.241 ± 0.173 | 0.785 ± 0.325 | **0.954 ± 0.008** | 0.956 ± 0.004 | 100.000 |
| BP-observation + Original | 0.216 ± 0.199 | 0.745 ± 0.366 | 0.914 ± 0.183 | 0.948 ± 0.093 | 98.000 |
| BP-obs + Original + Strategies | **0.310 ± 0.193** | **0.865 ± 0.229** | 0.937 ± 0.086 | **0.958 ± 0.015** | 100.000 |

Table 8: PPO performance on the Unlock-Pickup environment

| Configuration | R@500k | R@1.00m | R@2.50m | R@3.75m | % Solved |
|---|---|---|---|---|---|
| Original | 0.001 ± 0.001 | 0.000 ± 0.001 | 0.000 ± 0.001 | 0.000 ± 0.001 | 0.000 |
| Frame-Stack | 0.001 ± 0.003 | 0.000 ± 0.001 | 0.001 ± 0.002 | 0.001 ± 0.002 | 0.000 |
| RPPO with Original | 0.000 ± 0.001 | 0.000 ± 0.002 | 0.001 ± 0.002 | 0.000 ± 0.001 | 0.000 |
| Original + Strategies | 0.001 ± 0.002 | 0.001 ± 0.003 | 0.006 ± 0.046 | 0.014 ± 0.070 | 2.362 |
| BP-observation | 0.003 ± 0.008 | 0.028 ± 0.064 | 0.274 ± 0.408 | 0.383 ± 0.447 | 42.308 |
| BP-observation + Original | 0.006 ± 0.016 | 0.029 ± 0.071 | 0.286 ± 0.413 | 0.399 ± 0.459 | 42.308 |
| BP-obs + Original + Strategies | **0.011 ± 0.022** | **0.066 ± 0.133** | **0.410 ± 0.445** | **0.473 ± 0.460** | 52.174 |

Table 9: PPO performance on the Blocked-Unlock-Pickup environment

## E.2 −A2C

| Configuration | R@50k | R@150k | R@450k | R@700k | % Solved |
|---|---|---|---|---|---|
| Original | 0.753 ± 0.055 | 0.801 ± 0.022 | 0.859 ± 0.019 | 0.972 ± 0.019 | 100.000 |
| Frame-Stack | 0.765 ± 0.058 | 0.821 ± 0.032 | 0.807 ± 0.040 | 0.806 ± 0.037 | 46.667 |
| Original + Strategies | 0.753 ± 0.050 | 0.822 ± 0.025 | 0.862 ± 0.020 | 0.856 ± 0.022 | 86.667 |
| BP-observation | 0.889 ± 0.036 | **0.991 ± 0.001** | **0.992 ± 0.000** | 0.990 ± 0.004 | 100.000 |
| BP-observation + Original | 0.931 ± 0.019 | **0.991 ± 0.000** | **0.992 ± 0.000** | 0.990 ± 0.004 | 100.000 |
| BP-obs + Original + Strategies | **0.939 ± 0.016** | **0.991 ± 0.000** | **0.992 ± 0.000** | **0.992 ± 0.000** | 100.000 |

Table 10: A2C performance on the Empty 6x6 environment

| Configuration | R@300k | R@600k | R@1.50m | R@2.25m | % Solved |
|---|---|---|---|---|---|
| Original | 0.052 ± 0.015 | 0.059 ± 0.019 | 0.053 ± 0.019 | 0.058 ± 0.016 | 13.333 |
| Frame-Stack | 0.059 ± 0.023 | 0.061 ± 0.021 | 0.045 ± 0.020 | 0.063 ± 0.016 | 6.667 |
| Original + Strategies | 0.052 ± 0.016 | 0.046 ± 0.016 | 0.053 ± 0.015 | 0.058 ± 0.020 | 0.000 |
| BP-observation | 0.145 ± 0.063 | 0.406 ± 0.133 | **0.920 ± 0.037** | **0.971 ± 0.004** | 100.000 |
| BP-observation + Original | 0.139 ± 0.070 | 0.438 ± 0.086 | 0.913 ± 0.032 | 0.966 ± 0.003 | 100.000 |
| BP-obs + Original + Strategies | **0.178 ± 0.069** | **0.446 ± 0.095** | 0.911 ± 0.034 | 0.965 ± 0.003 | 100.000 |

Table 11: A2C performance on the Unlock environment

| Configuration | R@250k | R@500k | R@1.35m | R@2.00m | % Solved |
|---|---|---|---|---|---|
| Original | 0.042 ± 0.016 | 0.040 ± 0.015 | 0.047 ± 0.011 | 0.062 ± 0.020 | 0.000 |
| Frame-Stack | 0.047 ± 0.013 | 0.049 ± 0.020 | 0.049 ± 0.028 | 0.043 ± 0.018 | 0.000 |
| Original + Strategies | 0.044 ± 0.018 | 0.049 ± 0.014 | 0.058 ± 0.020 | 0.079 ± 0.048 | 0.000 |
| BP-observation | 0.136 ± 0.062 | 0.458 ± 0.106 | 0.957 ± 0.005 | 0.962 ± 0.002 | 100.000 |
| BP-observation + Original | 0.145 ± 0.037 | 0.496 ± 0.118 | 0.964 ± 0.009 | 0.973 ± 0.013 | 100.000 |
| BP-obs + Original + Strategies | **0.242 ± 0.107** | **0.656 ± 0.126** | **0.966 ± 0.007** | **0.976 ± 0.001** | 100.000 |

Table 12: A2C performance on the DoorKey 6x6 environment

| Configuration | R@600k | R@1.25m | R@3.20m | R@4.80m | % Solved |
|---|---|---|---|---|---|
| Original | 0.007 ± 0.005 | 0.007 ± 0.006 | 0.010 ± 0.010 | 0.015 ± 0.016 | 0.000 |
| Frame-Stack | 0.013 ± 0.018 | 0.009 ± 0.012 | 0.020 ± 0.034 | 0.029 ± 0.033 | 5.000 |
| Original + Strategies | 0.007 ± 0.005 | 0.008 ± 0.007 | 0.024 ± 0.025 | 0.072 ± 0.099 | 0.000 |
| BP-observation | 0.012 ± 0.020 | 0.041 ± 0.085 | 0.621 ± 0.208 | 0.865 ± 0.040 | 100.000 |
| BP-observation + Original | **0.021 ± 0.027** | **0.092 ± 0.087** | **0.739 ± 0.050** | **0.893 ± 0.031** | 100.000 |
| BP-obs + Original + Strategies | 0.020 ± 0.023 | 0.083 ± 0.080 | 0.730 ± 0.131 | 0.891 ± 0.043 | 100.000 |

Table 13: A2C performance on the DoorKey 8x8 environment

| Configuration | R@1.00m | R@2.00m | R@5.00m | R@7.50m | % Solved |
|---|---|---|---|---|---|
| Original | 0.002 ± 0.002 | 0.003 ± 0.004 | 0.005 ± 0.004 | 0.001 ± 0.003 | 0.000 |
| Frame-Stack | 0.003 ± 0.004 | 0.003 ± 0.003 | 0.004 ± 0.005 | 0.002 ± 0.003 | 0.000 |
| Original + Strategies | 0.005 ± 0.005 | 0.003 ± 0.006 | 0.004 ± 0.003 | 0.005 ± 0.006 | 0.000 |
| BP-observation | 0.007 ± 0.006 | 0.009 ± 0.008 | 0.146 ± 0.183 | 0.671 ± 0.219 | 85.000 |
| BP-observation + Original | **0.009 ± 0.007** | 0.018 ± 0.014 | **0.379 ± 0.263** | 0.814 ± 0.115 | 100.000 |
| BP-obs + Original + Strategies | 0.007 ± 0.007 | **0.025 ± 0.036** | 0.369 ± 0.248 | **0.824 ± 0.142** | 100.000 |

Table 14: A2C performance on the Unlock-Pickup environment

# F STATISTICAL SIGNIFICANCE OF PERFORMANCE DIFFERENCES

To assess whether performance differences between observation configurations were statistically significant, we conducted unpaired two-sample t-tests on the final episode rewards (averaged over the last 10 episodes). Each configuration was evaluated over approximately 30 independent training runs. We report significant differences at $p < 0.05$.

## F.1 PPO RESULTS

Across environments, PPO agents utilizing BP-derived representations generally achieved superior performance compared to the baselines. Notably, in the more challenging *Blocked-Unlock-Pickup* environment, the BP-observation configuration yielded substantial and statistically significant improvements compared to the original observation ($t = 6.952, p < 0.001$). In the *Unlock-Pickup* environment, the BP-observation configuration compared to the original observation did not yield statistically significant results ($t = 1.027, p = 0.305$), but when incorporating the original observation and strategies alongside BP-derived observations (BP-obs + Original and BP-obs + Original + Strategies), the differences became statistically significant ($t = 2.344, p < 0.021$ and $t = 3.158, p < 0.003$, respectively).

In simpler environments such as *DoorKey 6x6*, the BP-observation and Original configurations exhibited comparable performance ($t = 0.285, p = 0.777$). Incorporating strategies alongside BP-derived observations (BP-obs + Original + Strategies) led to a larger difference, though it did not reach statistical significance ($t = 1.010, p = 0.317$).

## F.2 A2C RESULTS

A2C agents also showed clear benefits from BP-derived information, particularly in the more complex environments. In all environments except *Empty 6x6*, baseline configurations without BP-based observations failed to converge, resulting in consistently low and statistically significant p-values when compared to BP-based configurations. For example, in the *Unlock* environment, BP-observation significantly outperformed the Original configuration ($t = 34.94, p < 0.001$), underscoring the impact of structured procedural information.

These results affirm the robustness of our approach and show its proven advantage in final performance, especially in more challenging environments.

# G HARDWARE

All experiments were conducted using an NVIDIA RTX 2080 GPU (8 GB VRAM).

# H BEHAVIORAL PROGRAMMING (BP) FORMALISM

This part summarizes the formal computational model underlying Behavioral Programming (BP), following the definitions introduced in the foundational literature (Harel et al., 2010; Elyasaf, 2021). BP models each behavior thread as a deterministic labeled transition system enriched with *requested* and *blocked* event sets. A system of b-threads is executed via an interleaving semantics obtained through a product construction over their respective transition systems. A behavioral execution mechanism selects, at each synchronization point, a single event that is requested and not blocked.

## H.1 DEFINITIONS

**Definition 1: Labeled Transition System.** A deterministic labeled transition system is a quadruple $\langle S, E, \rightarrow, init \rangle$, where

- $S$ is a set of states,
- $E$ is a set of events,
- $\rightarrow \subseteq S \times E \times S$ is a transition function,

- $init \in S$ is the initial state.

A run is a sequence $s_0 \xrightarrow{e_1} s_1 \xrightarrow{e_2} s_2 \xrightarrow{e_3} \cdots$, where $s_0 = init$, and for each $i$, the transition $s_{i-1} \xrightarrow{e_i} s_i$ is defined by $\rightarrow$.

**Definition 2: Behavior Threads.** Each behavior thread augments a deterministic transition system with annotations that indicate which events are *requested* and which are *blocked* in each state. More formally, a *behavior thread* (or *b-thread*) is a tuple $\langle S, E, \rightarrow, init, R, B \rangle$, where

- $\langle S, E, \rightarrow, init \rangle$ is a deterministic labeled transition system;
- $R : S \rightarrow 2^E$ assigns, to each state, the set of events *requested* by the b-thread when it is in that state;
- $B : S \rightarrow 2^E$ assigns the set of events *blocked* by the b-thread in that state.

A b-thread may also implicitly *wait for* events, by defining transitions for events not in $R(s)$.

**Definition 3: Run of a Set of B-Threads.** The semantics of a behavioral program, constructed of a set of b-threads, is obtained via a product construction, producing the set of all possible interlaced runs.

Given b-threads $\{\langle S_i, E_i, \rightarrow_i, init_i, R_i, B_i \rangle\}_{i=1}^n$, the run of the composed system are the runs of the labeled transition system $\langle S, E, \rightarrow, init \rangle$, where

$$S = S_1 \times \cdots \times S_n \quad , \quad E = \bigcup_{i=1}^{n} E_i \quad , \quad init = \langle init_1, \ldots, init_n \rangle.$$

The transition $\langle s_1, \ldots, s_n \rangle \xrightarrow{e} \langle s'_1, \ldots, s'_n \rangle$ exists if and only if:

$$e \in \bigcup_{i=1}^{n} R_i(s_i) \quad \text{(the event is requested)}, \tag{1}$$

$$e \notin \bigcup_{i=1}^{n} B_i(s_i) \quad \text{(the event is not blocked)}, \tag{2}$$

and for every $i$,

$$(e \in E_i \implies s_i \xrightarrow{e}_i s'_i) \quad \text{(affected threads move)}, \tag{3}$$

$$(e \notin E_i \implies s_i = s'_i) \quad \text{(unaffected threads stay put)}. \tag{4}$$

## H.2 Example Under the Formal Semantics

We now visit a simplified *Key–Door–Goal* scenario in terms of the above definitions. The possible system events are: $E = \{AtKey, PickUpKey, OpenDoor, ReachGoal\}$.

We specify a behavioral program consisting of four b-threads, each is defined as a tuple $\langle S_i, E, \rightarrow_i, init_i, R_i, B_i \rangle$ (see Definition 2). The synchronization points (sync) are represented as states in the LTS, where each state specifies the set of events it requests ($R$) and blocks ($B$). The events that are waited for are specified on the edges.

```
B-Thread MoveToKey:
```

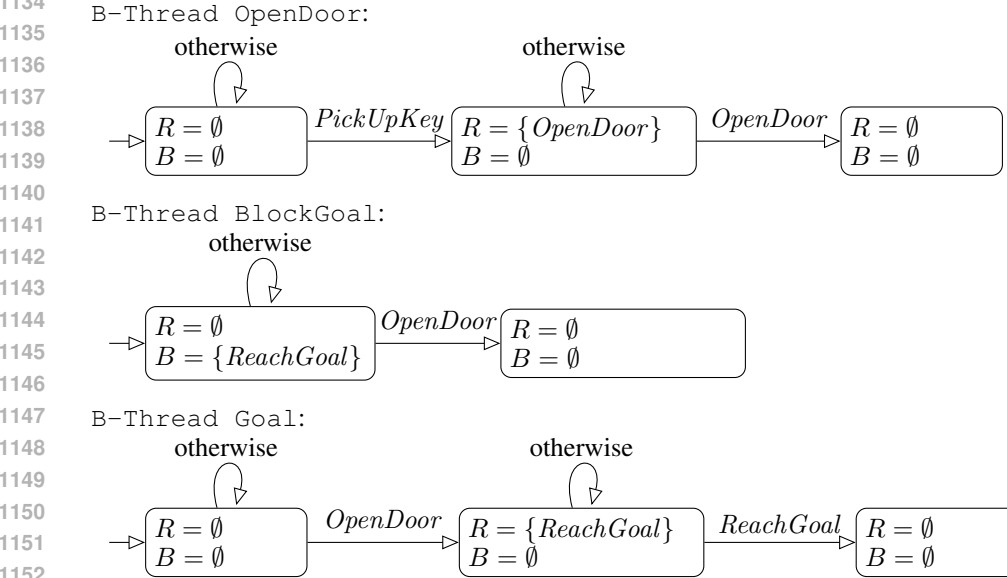

**Execution of the Behavioral Program.** We illustrate the execution of the behavioral program using the semantics outlined in Definition 3.

Initially, all b-threads are in their respective start states. When the agent arrives at the key, the $AtKey$ event is triggered. Consequently, the `MoveToKey` b-thread transitions to its next state, where it requests the $PickUpKey$ event. Upon the selection of this event, the b-thread advances to its final state and terminates. Concurrently, the `OpenDoor` b-thread also advances, as it is waiting for the $PickUpKey$ event. This execution sequence yields the following trace: $AtKey, PickUpKey, OpenDoor, ReachGoal$.

The blocking statement in the `BlockGoal` b-thread ensures that the $ReachGoal$ event cannot be selected before the door has been opened.

In summary, BP provides a principled, compositional mechanism for interleaving independent behavioral requirements via event-based synchronization. Crucially, the underlying rigorous mathematical semantics facilitate the formal verification of the specification. Although verification is beyond the scope of this work, it is a fundamental strength of the paradigm and has been extensively studied (Harel et al., 2013).

## I  EXAMPLE BP-STRATEGIES AND THEIR INTEGRATION

We recall that these strategies represent a BP-based manual feature engineering, allowing a domain expert to handcraft features based on their knowledge. This contrasts with our core proposal, where a complete and semantically rich state representation is derived fully and automatically from the environment's BP specification.

In the paper, we provide three types of these strategies:

1. A simple **turn-counting** strategy.

2. A **phase-level** strategy for identifying the high-level progress of the task.

3. A **distance-to-objective** strategy providing distance from next objective (e.g., key, door...).

Each strategy is implemented using BP. The strategy representation is a $H \times W$ matrix filled with a scalar value. This representation is concatenated to the original observation (also of dimension $H \times W$). We illustrate two of these strategies in the following examples to provide greater clarity to the reader.

## I.1  MINIMAL EXAMPLE: COUNTING CONSECUTIVE TURNS

The following strategy b-thread monitors agent actions and exposes the number of consecutive left turns. This information does not appear in the original MiniGrid observation and serves as a lightweight behavioral feature:

```
@b_thread
def count_left_turns_bt(bt_obs):
  turns_left = 0
  while True:
    bt_obs.update_observation(turns_left)
    e = yield sync(waitFor=any_agent_action)
    if e == ACTIONS["left"]:
      turns_left += 1
    else:
      turns_left = 0
```

This thread maintains a small internal state and updates its symbolic observation after every agent action. That is, the $H \times W$ will be filled with the turns_left number.

## I.2  TASK-PHASE STRATEGY

For more complex environments (i.e., Unlock, UnlockPickup, BlockedUnlockPickup, DoorKey), we implement a *phase-level* strategy. This strategy encodes intuitive symbolic stages of the task:

```
@b_thread
def unlock_env_level_bt(bt_obs):
  while True:
    bt_obs.update_observation(0)                # initial phase
    yield sync(waitFor=picked_up_key)
    bt_obs.update_observation(1)                # key acquired
    e = yield sync(waitFor=[dropped_key, unlocked_doorevent])
    if e == unlocked_door:
      bt_obs.update_observation(2)         # door unlocked
```

This strategy captures the intuitive progression humans employ when mentally solving the task.

