# OpenReview forum: "BPRL: A Behavioral Approach to State Representation in Reinforcement Learning"
_ICLR.cc/2026/Conference — Submitted to ICLR 2026_

### Official Review · Reviewer_V7yu · 2025-10-24

**Soundness:** 1
**Presentation:** 1
**Contribution:** 2
**Rating:** 2
**Confidence:** 4

**Summary:**

The authors present BPRL, a framework for integrating various tools from Behavioral Programming (BP) into Deep RL agents. The authors first present a method for distilling environments represented using BP into something usable by a learning agent by virtue of programmatically providing state observations that are not immediately derivable from the BP environment logic. The authors then compare agents trained on these observations to multiple baselines, including a set with image-based observations and additional “BP-strategies” that contain heuristics for environment solutions.

1. **What is the specific question and/or problem tackled by the paper?**

    The problem of training DRL agents on environments specified by Behavioral Programs.

2. **Is the approach well motivated, including being well-placed in the literature?**

    The first part of the paper describing a method for deriving state observations from a BP environment is sensible, but I don’t believe the second half of the paper, on integrating BP-Strategies is well-placed in the literature.

3. **Does the paper support the claims? This includes determining if results, whether theoretical or empirical, are correct and if they are scientifically rigorous.**

    It’s unclear whether the results meaningfully and convincingly support the claims made in the paper. Specifically, there are issues with the third claim the authors make:

    > We empirically demonstrate that BPRL significantly improves learning efficiency and performance across multiple RL algorithms and environments, addressing key challenges in scaling RL to complex, high-dimensional tasks.
    >

    IIUC, the authors made a slightly unusual choice to not do a parameter sweep over each agent configuration, so agents with potentially different-sized observation spaces will use the same hyperparameters. It is possible that agent performance would be different after a parameter sweep, which would contradict the claim made by the authors. Furthermore, it is not clear that “key challenges in scaling RL to complex, high-dimensional tasks” are genuinely addressed.

4. **What is the significance of the work? Does it contribute new knowledge and sufficient value to the community?**

    It potentially contributes new knowledge to the Behavioral Programming and BP-DRL community, though the value of its contribution is unclear.

**Strengths:**

The work is connected nicely to the field of Behavioral Programming, the wording and explanations of things are general clear and easy to follow, parts of the work are original (especially WRT deriving states from BP environment specifications).

**Weaknesses:**

The contribution of the work is only somewhat significant, the paper meanders between two different topics: getting state observations from BP environments and specifying advice using BP-strategies, the results are not particularly compelling or well-described/presented, and the work is poorly situated in the broader field of leveraging symbolic knowledge in DRL.

The paper is trying to do too many things at once, and does each of them poorly. There are significant methodological and interpretability issues, from making hyperparameter choices that may obfuscate results to making apples to oranges comparisons in reward tables. Finally, the work is extremely poorly situated in the broader RL community outside of the BP niche.

**Questions:**

The differences in performance between various agents are quite opaque in table 1 especially for the PPO agents. It’s not immediately clear that there are substantial differences in performance. Furthermore the table ignores the temporal aspect of learning by merely reporting the final rewards achieved in environments.

There are potentially too many things in this table, I’m not sure it’s useful to compare all these things. The comparison between original observation, frame-stacked observation, and the BP-derived observation and BP-derived observation + original is sensible, since these are designed to essentially provide ground-truth information about the underlying world state. However, the inclusion of the two BP-strategies agent in this table is unusual. Comparing these agents means something very different, since the BP-strategies agent contains additional heuristics. It’s genuinely hard to make heads or tails of what is useful here to compare, it’s currently comparing apples to apples to oranges to guavas.

Examples of the BP-Strategies (in full code) in the main paper would be useful. At the very least in the appendix, but there are none.

The discussion on Neuro-symbolic RL is seriously lacking, with only a two-sentence paragraph making vague reference to existing methods which integrate symbolic reasoning into DRL. There are numerous structured languages used to provide supplemental or total advice about decision processes, e.g. LTL, PDDL, RDDL, RLang, Policy Sketches (from Andreas et al. 2017) just to name a few, and there are even more works which leverage those languages in learning agents. This work is incomplete without making reference to both the languages used by the community and the works that integrate those languages into learning.

---

> ### Author Response · Authors · 2025-11-20
> **Authors’ Response to Reviewer Comments**
>
> We thank the reviewer for their critical feedback. We acknowledge the reviewer's concerns, which have led to low scores for Presentation and Soundness. We believe these concerns, particularly the view that the paper "meanders between two different topics" and makes "apples to oranges comparisons", stem from a fundamental misunderstanding of our paper's experimental design, which we apologize for not making clearer.
>
> ## On the "Meandering" Structure and "Apples to Oranges" Comparisons
> The reviewer's core critique is that the paper is split between two distinct topics: (1) automatic state derivation from BP environments (Section 3.2)  and (2) manual advice via 'BP-strategies' (Section 3.3). We must clarify this structure: **(1) is our core contribution, and (2) is our primary ablation baseline.** The 'BP-strategies' (Section 3.3)  are not a separate, muddled contribution. They are our explicit implementation of manual feature engineering. We introduced this *Original + strategies* baseline precisely to answer the question: "Is our automatic $o_{BP}$ representation (Section 3.2)  better than hand-crafted, manual feature engineering?". Therefore, the comparison in Table 1 is not *"apples to oranges"*. It is a direct, empirical comparison between our automatic method (BP-observation + Original) and a strong manual alternative (Original + strategies). The results, which show our automatic configuration consistently outperforms this manual baseline (as seen in Tables 1, 2 and Figure 3), provide strong validation for our core contribution.
>
> ## On Results and Temporal Analysis
> While Table 1 reports final rewards, the temporal and sample-efficiency results
> appear in Figure 3 and Table 2. These clearly show our BP-based methods
> converging faster and to higher returns, especially in long-horizon tasks such as
> Blocked-Unlock-Pickup. The reviewer’s concern about final rewards is valid, but
> the differences are substantial in A2C and visible in both Table 1 and the Appendix
> Figure 4.
>
> ## On Hyperparameters
> Our hyperparameter tuning appears to have been misunderstood. As detailed in Appendix~C, tuning was conducted extensively but *only* on the 'Original' baseline. The same optimized values were then applied to all configurations for fairness. This likely disadvantaged the BP-based settings, as they were not tuned for their observation spaces. We did perform additional sweeps, which confirmed that these parameters were generally effective across configurations, though these details were omitted for brevity. The strong improvements we observe, despite this baseline-focused tuning, further support that our results are not an artifact of hyperparameter choices.
>
> ## On Positioning Within the Literature
> We agree that the discussion of neuro-symbolic RL (Section~5.2) is too brief. We will expand it to contextualize BPRL relative to LTL, PDDL, RLang, and related formalisms. Prior BP-RL work primarily uses BP for policy supervision, planning, or reward shaping, similarly to LTL/PDDL; our novelty lies in using BP's formal semantics for *automatic observation construction*. We will revise this section to better situate our representation-focused contribution within the broader field.
>
> ## On Missing Code
> This is a valid omission. We will add code examples for the 'BP-strategies' (Section 3.3) to the appendix to improve clarity and reproducibility. The code is available anonymously through the link in the paper, but the reviewer's point is correct, and a few code examples in the appendix may clarify some of the misunderstandings regarding the 'BP-strategies'.
>
> ## On Scaling
> Finally, regarding the claim of addressing "key challenges in scaling", this refers to our framework's development scalability (i.e., "non-intrusive extensibility") and its ability to scale to complex tasks. For our discussion on scaling to partially observable and continuous domains, we respectfully refer the reviewer to our general response, which addresses these points in detail.

---

> > ### Comment · Reviewer_V7yu · 2025-11-25
> >
> > Thank you for clarifying the ablation runs, but this raises another question:
> >
> > How were the manually-defined strategies chosen? If they *could have* been chosen intentionally to be weaker than the automatic observations, why are they a reasonable point of comparison? The specific choice of strategies should be defended in order to justify the claims made by this comparison (that automatic BP observations are better than manually-defined strategies).
> >
> > Relating this work to prior work is critical — without it, the work feels solipsistic — please revise this section and I will consider it in my final score. Likewise, please include code examples for the BP-strategies in the appendix, and I will consider it in my final score.

---

> > > ### Author Response · Authors · 2025-11-26
> > > **Response to Follow-up: Strategy Selection, Related Work, and Code**
> > >
> > > Thank you for your valuable feedback. We appreciate the opportunity to clarify the motivation behind the manually-defined strategies and to strengthen the paper's context within prior work.
> > >
> > > We have expanded the discussion on the selection of baseline strategies in the revised manuscript. Additionally, as requested, we have included two concrete code examples of these BP-strategies in the Appendix to better illustrate their implementation and complexity.
> > >
> > > To directly address your question on whether these strategies were chosen to be "weak": they were not chosen to be weak, but rather to be representative of a comparable level of human effort.
> > >
> > > The core value proposition of our framework is the "natural alignment" between the code and the requirements. In our approach (BP-observation), the b-threads are essentially a formal, executable representation of the requirement text. To create the BP-observation, the user simply specifies the environment's requirements and translates them into BP; the state representation is then derived automatically.
> > >
> > > To make a fair comparison, we sought a baseline that reflects a similar workflow:
> > > 1.  **BP-Observation:** User specifies environment mechanics -> Translates to BP -> Agent gets state.
> > > 2.  **BP-Strategies:** User identifies simple behavioral heuristics -> Translates to BP -> Agent gets state.
> > >
> > > While it is certainly possible to engineer highly sophisticated heuristics that might achieve higher performance, doing so would represent a significantly higher degree of human engineering and domain expertise than what is required for the BP-observation method. We purposefully selected "natural behavioral cues", such as counting consecutive left/right turns, identifying the task phase (subgoals), and tracking distance, because these represent the kind of simple, intuitive heuristics a user would naturally reach for without engaging in complex feature engineering.
> > >
> > > Regarding prior work, we agree that contextualizing this work is critical. We have significantly revised the "Related Work" section to better situate our contributions within the broader landscape of state representation learning and neuro-symbolic RL, ensuring the work is properly anchored in the existing literature.

---

> > > > ### Comment · Reviewer_V7yu · 2025-11-26
> > > >
> > > > Thank you for addressing some of my concerns, but I still do not believe this work to be worthy of acceptance to ICLR. The comparison of BP-Observation to BP-Strategies doesn't say much if the line drawn by the authors in crafting the BP-Strategies is arbitrary ("a comparable level of human effort"). Including this as a basis of comparison truly requires more justification.
> > > >
> > > > The updated Related Works section still reads extremely thin. There is a tremendous amount of research that leverages formal languages to augment policies and state observations, and a very small amount of it is referenced in this section despite there being free room on the final page.
> > > >
> > > > Accordingly, I will raise my score, though it will still be below the acceptance threshold.

---

### Official Review · Reviewer_Jpnj · 2025-10-26

**Soundness:** 3
**Presentation:** 2
**Contribution:** 2
**Rating:** 4
**Confidence:** 4

**Summary:**

The paper presents a novel framework for modeling environment dynamics in reinforcement learning (RL), centered around the use of Behavioral Programming (BP). BP is a paradigm for constructing reactive systems by specifying behavioral threads that define what the system may, must, or must not do. This approach enables a modular, rule-based encoding of the environment's behavior.

A key contribution of the work is an automated method for deriving a structured state representation of the environment. As I understand it, each behavioral rule is compiled into a deterministic automaton (called LTS), and the overall environment state is formed by concatenating the current states of these individual automatons.

In addition to modeling the environment, the paper demonstrates how BP can be used to inject prior knowledge into the RL agent. These BP rules do not need to fully specify the environment's dynamics; instead, they can encode partial knowledge that the designer considers useful for the agent. Each rule is again compiled into an automaton, and the agent's state is augmented with the current state of these automatons, effectively integrating domain knowledge into the learning process.

The framework is evaluated in the MiniGrid environment, where the features derived from the BP-based representation outperform the standard baseline features, which are typically based on the agent’s raw RGB observations.

**Strengths:**

This is a clear and well-written paper that successfully bridges reinforcement learning (RL) and the Behavioral Programming (BP) literature. The work is technically sound and introduces a novel approach to encoding environment dynamics in RL. While the effectiveness of BP as a general-purpose tool for modeling such dynamics may be open to debate, the idea of using a formal language from which state representations can be automatically extracted is promising. This could be particularly beneficial for new RL practitioners, who often struggle to define a suitable Markovian state space for their problems. In the long term, I see potential in learning behavioral threads (b-threads) from experience, which could help RL agents better manage partial observability and improve generalization.

**Weaknesses:**

**Critical Comments**

The paper does not discuss why the BP-derived observations outperform the original observations in MiniGrid. I suspect the core reason is that BP-based observations contain _privileged information_ that the agent ideally should not have access to. A key aspect of MiniGrid is its partial observability: the agent can only perceive its immediate surroundings, which makes it a realistic benchmark for RL agents operating under uncertainty.

By contrast, BP-derived observations appear to remove this constraint. Although the paper does not explicitly detail what information is included in the BP-based state representation, the method's description suggests it likely includes comprehensive environment details—such as the agent’s exact location, the status of doors (open or closed), and the positions of keys outside the agent’s field of view. This effectively transforms the problem into a fully observable one, making the task significantly easier and less realistic. For example, a robot trained in simulation using BP-derived observations would not have access to such privileged information when deployed in the real world.

This is a crucial point that is currently missing from the paper and deserves further discussion.

---

**Conceptual Concerns**

The paper also does not address a core challenge in RL. Once the full environment dynamics are programmed, deriving a proper state representation is relatively straightforward—any Markovian state would suffice. The only scenario in which this becomes difficult is when the designer lacks a basic understanding of RL principles.

One could argue that some Markovian states are more informative than others, and perhaps BP-derived features are optimal in some sense. However, the paper provides no evidence to support this claim. The only comparison made is between BP-derived features (which are Markovian) and raw RGB observations (which are intentionally non-Markovian to preserve realism in MiniGrid). In such a comparison, it is unsurprising that the Markovian features perform better.

A more compelling evaluation would compare BP-derived features against alternative Markovian representations, or formally demonstrate that BP-derived features are superior in general.

---

**Minor Concerns**

- **Section 3.1** states: "_As is standard in DRL, manual implementation of the environment is a prerequisite_." This is not universally true. Sometimes, RL agents learn directly from interaction with the physical environment. Moreover, in most simulation-based research, the environment is already implemented—whether it's a physics engine or a video game—so manual implementation is not always required.

- **Limitations of BPRL**: The paper does not discuss the limitations of the proposed approach. In particular, BPRL may not be suitable for modeling continuous domains, which are common in robotics and other impactful RL applications. Since BP relies on labeled transition systems (LTS), representing continuous variables would require an infinite number of states. Although the paper briefly mentions extending BPRL to continuous domains as future work, it does not address the implications or challenges of doing so.

- **Formal Languages in Reinforcement Learning**: Consider including a discussion of prior work that has incorporated formal languages into reinforcement learning beyond BP [e.g., 1-6].

[1] Littman, M. L., Topcu, U., Fu, J., Isbell, C., Wen, M., & MacGlashan, J. (2017). Environment-independent task specifications via GLTL. arXiv preprint arXiv:1704.04341.

[2] Icarte, R. T., Klassen, T., Valenzano, R., & McIlraith, S. (2018). Using reward machines for high-level task specification and decomposition in reinforcement learning. In International Conference on Machine Learning (pp. 2107-2116). PMLR.

[3] De Giacomo, G., Iocchi, L., Favorito, M., & Patrizi, F. (2018). Reinforcement learning for LTLf/LDLf goals. arXiv preprint arXiv:1807.06333.

[4] Jothimurugan, K., Alur, R., & Bastani, O. (2019). A composable specification language for reinforcement learning tasks. Advances in Neural Information Processing Systems, 32.

[5] Vaezipoor, P., Li, A. C., Icarte, R. A. T., & Mcilraith, S. A. (2021). Ltl2action: Generalizing ltl instructions for multi-task rl. In International Conference on Machine Learning (pp. 10497-10508). PMLR.

[6] Yalcinkaya, B., Lauffer, N., Vazquez-Chanlatte, M., & Seshia, S. (2024). Compositional automata embeddings for goal-conditioned reinforcement learning. Advances in Neural Information Processing Systems, 37, 72933-72963.

**Questions:**

1. What specific information is included in the BP-derived observations, and why do these features lead to better performance compared to the original MiniGrid features?

2. Do the BP-derived observations guarantee Markovian properties with respect to the environment's transition dynamics?

3. Is it feasible to extend the BPRL framework to continuous domains? Given that Behavioral Programming relies on discrete labeled transition systems, it is unclear how it would handle continuous variables without requiring an infinite number of states. Could the authors elaborate on the challenges and potential solutions?

4. Are the BP-derived features optimal in any formal or empirical sense? Has any analysis been conducted to evaluate whether these features offer advantages over other Markovian representations, either theoretically or through comparative experiments?

---

> ### Author Response · Authors · 2025-11-20
> **Authors’ Response to Reviewer Comments**
>
> We thank the reviewer for their detailed and valuable feedback. We are grateful for the positive assessment of our paper as "technically sound" and for recognizing the novelty of using a formal language to automatically extract state representations. The reviewer's primary critiques-regarding "privileged information", a "non-Markovian" baseline, and the "straightforward" nature of deriving a state - all appear to stem from a single, critical misunderstanding of our experimental baseline, which we acknowledge our presentation (rated "fair") may have obscured. We are confident that clarifying this point will resolve these conceptual concerns.
>
> ## Q1, Q2, and Weaknesses
> A concern was raised that our BPRL-derived observation ($o_{BP}$) provides an unfair advantage (*"I suspect the core reason is that BP-based observations contain privileged information"*). We must clarify that this is a misunderstanding of our baseline. As defined in our Baselines (Section 4.1), our 'Original' observation is *not* the standard partially-observable agent view. Instead, to provide a much stronger baseline, we use the **fully observable** $H \times W \times 3$ grid, which encodes the entire environment's contents. Thus, both our BPRL agent and our baseline agent receive the *exact same* global information.
>
> This clarifies two points: (1) Both representations are fully Markovian (answering Q2). (2) The superior performance of BPRL (Figure 3, Tables 1 and 2) (answering Q1) is therefore not due to receiving more information, but from receiving a superior **representation** of that same information. Further details of this point can be found in the official comment posted for all of the paper reviewers.
>
>
> ## Conceptual Weaknesses and Q4
> * The reviewer argues that deriving a state is "straightforward" once dynamics are programmed (W-Conceptual 1). We respectfully argue this confuses two distinct, manual tasks: (1) Environment Modeling and (2) Feature Engineering. While standard DRL requires manual effort for both, our framework's core novelty is that the manual effort for (1) Environment Modeling (Section 3.1) allows (2) Feature Engineering to be performed **100\% automatically** (Section 3.2). This eliminates the second manual step.
> * W-Conceptual 2 and Q4 ask for a comparison against "alternative Markovian representations". Our paper provides this comparison. As noted in our general response, our 'Original' baseline is the fully observable, Markovian grid, not non-Markovian RGB. Thus, our main results (Tables 1-2, Figure 3)  already compare our automatic representation ($o_{BP}$) against a standard, high-level Markovian one ('Original').
>
>     Furthermore, to *directly* answer Q4, we introduced the 'BP-strategies' (Section 3.3) to serve as an *additional* baseline. This *Original + strategies* configuration represents a **manually-designed**, high-level, alternative Markovian representation. Our results (Tables 1-2, Figure 3)  show that our **automatic** $o_{BP}$ generally **outperforms** this **manual** baseline, providing the precise empirical validation requested.
>
>     While we do not claim that $o_{BP}$ is *formally* optimal, these comparisons demonstrate that it is a **practically strong** and **consistently superior** representation relative to realistic manual alternatives.
>
> ## On Minor Concerns (W-Minor 1, W-Minor 3)
> We agree with W-Minor 1 that "manual implementation" (Section 3.1) is not universally true (e.g., physical robots) and will revise our text to clarify this applies to new simulation-based development. Finally, we thank the reviewer for the excellent and highly relevant list of references [1-6] (W-Minor 3). We will gladly incorporate them into our Related Work (Section 5) to better situate BPRL in this landscape. Our initial selection of related work was necessarily concise due to strict page limitations and our focus on the most directly related methods; however, we agree that these additions provide valuable context and will strengthen the paper.
>
>
> ## Minor concern 2 and Q3
> We addressed these concerns in our official comment, posted for all reviewers of the paper.

---

### Official Review · Reviewer_RXZR · 2025-10-30

**Soundness:** 2
**Presentation:** 1
**Contribution:** 2
**Rating:** 4
**Confidence:** 4

**Summary:**

This paper focuses on state representation learning for deep reinforcement learning. It identifies a key challenge in existing DRL methods: the lack of compact, time-aware, semantically structured state representations. To tackle these problems, this paper proposes BPRL, which leverages Behavioral Programming to implement environments and automatically derive a state representation for RL. Each behavioral thread is treated as a labeled transition system; at every step, the set of current synchronization states across threads is encoded and concatenated into a fixed-size representation. This representation can be fused with raw inputs in a dual-branch policy network. Experiments on MiniGrid demonstrate that BPRL accelerates learning, improves final returns, and enhances sample efficiency over non-BP baselines.

**Strengths:**

1. The paper is well-organized and easy to follow, presenting clear code listings and a step-by-step construction of b-threads that make the proposed method concrete and reproducible.

2. Similar to rule-based approaches, the behavioral programming method in this paper offers greater interpretability than learning-based state representation methods.

3. This paper also demonstrates the effectiveness of the method, achieving substantially higher sample efficiency, as shown in Figure 3.

**Weaknesses:**

1. My primary concern lies in the limited scalability and applicability of this paper. The BP-based methods, such as the b-threads illustrated in Listings 1 and 2, appear to rely on prior knowledge of the environment’s operational rules and still have the need for manual feature engineering. Consequently, it may be difficult to extend these methods to more complex or unknown environments. In my view, the b-threads resemble logic-based rules that function as language-like instructions. While this design enhances interpretability, it does not embody a learning-driven approach and therefore struggles to generalize to diverse benchmarks where agents lack prior knowledge of the environments.

2. Moreover, this paper do not compare with other popular learning-based state representations such as bisimulation metric and BP-based methods mentioned in the related work. And all experiments are only conducted on the MiniGrid benchmark, with the original observation being a fully observable grid. This leaves open the method’s scalability to richer visual inputs, partial observability, and continuous-control benchmarks.

3. The overall quality of writing is relatively weak. For example, the term non-intrusive extensibility mentioned in Line 63 is unclear and requires further explanation. Similarly, the repeated use of modular and incremental is not well defined or adequately elaborated within the text.

**Questions:**

1. When the rules are unclear or relatively complex, how should we write b-threads? For example, in continuous control (e.g., DMControl or MuJoCo), how do we map continuous trajectories and actions into discrete events and how to design synchronization points?

2. BP code seems to be environment-specific. Is there a more automated and generalizable way to write BP code? For example, Section 3.1 mentions that LLMs can translate requirements into BP code, could you provide a concrete workflow?

3. What exactly does the term multi-modal state representation in Line 53 refer to? Does the author provide any corresponding experiments or empirical results to validate this concept?

---

> ### Author Response · Authors · 2025-11-20
> **Authors’ Response to Reviewer Comments**
>
> We thank the reviewer for their constructive and professional feedback. The primary concerns regarding manual effort, scalability, and comparisons appear to reflect a difference in interpretation of our work's core novelty, which we acknowledge our presentation (rated "poor") may have obscured.
>
> ## W1
> The reviewer raised an important point that allows us to clarify our core contribution. The reviewer correctly identifies that writing b-threads (Listings 1-2) requires manual effort; however, this effort is for environment modeling (Section 3.1), a prerequisite step in any RL setup.
>
> Our framework's novelty (Section 3.2, "Automatic State Representation") is that it eliminates the subsequent, separate manual step of feature engineering. It does this by automatically deriving the state vector $o_{BP}$ directly from the formal semantics of that same environment model. The agent remains fully "learning-driven"; BPRL simply provides it with a superior, structured observation, which our results show dramatically improves learning (Figure 3, Tables 1-2).
>
> ## W2, Q1
> Regarding comparisons (W2): Prior BP-RL integrations (discussed in Section 5.2) used BP for policy supervision (e.g., shields) or reward shaping. BPRL is the first framework to shift BP's role to observation construction, using the model's internal state as the observation itself. Methods like bisimulation are complementary, not directly competitive; they learn an abstraction from data, while we derive one from a formal specification. The concerns about the MiniGrid benchmark, partial observability, and extensibility to continuous domains are addressed in our general response.
>
> ## W3
> We apologize for the lack of clarity regarding key BP terms. The terms "modular",  "incremental",  and "non-intrusive extensibility" (Line 63) are defined and demonstrated concretely in our general response, which tries to clarify further the scalable development process shown in Listings 1 and 2. We have addressed this issue in the revised version. For completeness, we clarify here that “non-intrusive extensibility” refers to BP’s ability to extend a system by adding new, independent b-threads without modifying existing ones - a property demonstrated concretely in Listings 1-2.
>
> ## Q2
>  The reviewer's intuition is correct. This is a highly promising research area. Our brief mention (Section 3.1) refers to concrete, published work (e.g., Yaacov et al., 2024; Harel et al., 2024) demonstrating that LLMs can effectively translate natural language requirements directly into executable BP code.
>
> The general workflow is as follows. Instead of asking an LLM to generate a complex, monolithic system from scratch, the LLM is prompted to translate each high-level, natural language requirement into its own dedicated code module (a b-thread). These modules can specify both desired behaviors (scenarios) and forbidden behaviors (anti-scenarios). An application-agnostic BP execution engine then interprets and interweaves all these small, independent modules at runtime to produce the final, cohesive system behavior that satisfies all the original requirements. This requirement-per-module approach not only "harmonizes well with the capabilities of LLMs" but also simplifies the verification of the generated code.
>
> ## Q3
> The term "multi-modal state representation" (Line 53) (Q3) is detailed in our general response. It refers to the fusion of the raw visual input and our symbolic $o_{BP}$ vector via the dual-branch architecture (Section 4.2), and is empirically validated by our BP-observation and the original results (Figure 3, Table 1).

---

### Official Review · Reviewer_PEux · 2025-11-03

**Soundness:** 3
**Presentation:** 2
**Contribution:** 3
**Rating:** 4
**Confidence:** 5

**Summary:**

This paper explores the use of Behavioral Programming (BP), a scenario-based modeling paradigm, for constructing structured environment representations in Deep Reinforcement Learning (DRL). The key idea is that the same BP model that specifies the environment can also provide semantically rich state representations for RL agents. The authors test this idea on MiniGrid benchmarks, comparing PPO and A2C agents trained on raw, frame-stacked, and BP-derived observations. They report that BP-based representations yield higher sample efficiency and faster convergence.

The paper is well-intentioned and attempts to connect the formal methods community with DRL by introducing structured modeling principles into representation learning.

**Strengths:**

- The idea of leveraging a formal modeling framework (Behavioral Programming) to generate structured representations for DRL opens an interesting interdisciplinary direction.
- Experiments show meaningful gains in sample efficiency and learning speed, suggesting that structured representations indeed help DRL performance.
- The paper correctly highlights the mismatch between real-world structured systems and unstructured end-to-end learning, motivating the need for scenario-based abstractions.
- BP’s event-based structure could provide interpretability benefits, though this aspect is not deeply explored.

**Weaknesses:**

- The paper does not adequately justify why BP is a particularly suitable modeling formalism for RL. Other structured paradigms such as Hierarchical RL, Options, Programmatic RL, or Recursive RL address similar goals, yet the paper neither contrasts nor situates BP within this landscape. This makes the contribution feel incremental or somewhat arbitrary.
- The technical section assumes significant prior knowledge of BP concepts (e.g., b-threads, synchronization points, request/block/wait semantics). Without a concise self-contained introduction, the paper risks alienating much of the ICLR audience unfamiliar with this formalism.
- It is not well explained what the RL agent is optimizing in a BP-based environment (reward modeling), or how stochasticity and nondeterminism in BP models are handled.
- All experiments are conducted on MiniGrid, a relatively simple benchmark. It is not clear whether the evaluated environments involved any stochasticity in their transition dynamics or were entirely deterministic.
- The paper could be strengthened by an analytical discussion of *why* BP-based representations help (e.g., by measuring reduction in state entropy, improved temporal abstraction, or compositional generalization). Without this, the results, though promising, remain somewhat anecdotal.

**Questions:**

- How does the system handle stochastic environments or partial observability?
- Could BP representations be automatically inferred, or are they manually designed?
- What are the expressiveness or scalability limits of BP compared to hierarchical, programmatic, or recurisve RL?
- Can the proposed method generalize beyond MiniGrid to more realistic or continuous-control settings?

**Details Of Ethics Concerns:**

None.

---

> ### Author Response · Authors · 2025-11-20
> **Detailed Response to Reviewers W1–W5 and Questions Q1–Q4**
>
> We sincerely thank the reviewer for their constructive feedback.
>
> ## W1, Q3
> We appreciate the comparison to HRL, Options, and Programmatic RL, as it allows us to clarify BPRL's unique position.
> These frameworks provide temporal abstraction of the policy (decomposing $\pi(a|s)$) across time scales. BPRL addresses an orthogonal problem: automated behavioral abstraction of the state representation ($s$). We do not modify the policy architecture; instead, we provide a novel method to automatically derive semantically rich, structured state vectors $O_{BP}$ directly from the formal BP environment model (Section 3.2), thereby eliminating manual feature engineering while capturing temporal dependencies and symbolic structure. This makes BPRL **complementary to, not competitive with**, hierarchical approaches.
>
> Regarding expressiveness (Q3): BP's modularity enables incremental composition of complex behaviors without refactoring (Listings 1-2), and the state dimensionality scales linearly with the number of behavioral threads. Unlike hierarchical decompositions that require pre-defined goal structures, BP automatically exposes procedural semantics through synchronization points, which our framework encodes without additional programming effort.
>
>
>
> ## W2, W5
>
> * We recognize that BP is a formalism that may be unfamiliar to the broader ICLR audience, and that Section 2.1 could be dense for those readers. To support accessibility, Section 2.1 does provide a self-contained introduction to BP concepts (b-threads, synchronization points, request/block/wait semantics) with concrete MiniGrid examples (Listings 1-2). Additionally, we have added an Appendix with an extended tutorial-style overview of the formalism, and we will improve signposting in the revision to guide readers to these resources.
>
> * Regarding the analytical justification (W5), Section 3.2 explains how $O_{BP}$ captures (i) temporal structure (event progression within behaviors) and (ii) behavioral intent (functional goals encoded in b-threads). These properties directly address the common problem of *state aliasing*, where different states are mapped into the same state encodings (which is problematic, assuming they have different optimal actions). Moreover, our compositional structure enables better generalization across behavioral phases (e.g., "searching for key" vs. "unlocking door").
>
>    Our empirical results (Tables 1-2, Figure 3) demonstrate this translates to significantly improved sample efficiency and convergence - agents’ mean reward **increased by 230–600\%** after the same number of steps, indicating more efficient exploration of the structured state space. We will expand the discussion to explicitly connect BP's structural properties to these learning advantages.
>
> ## Q2
>
> Regarding Q2 (automatic vs. manual): The BPRL state representation $O_{BP}$ is fully automatic. As described in Section 3.2 ("Automatic State Representation"), $O_{BP}$ is algorithmically extracted from the BP model's formal semantics - specifically, from each b-thread's current synchronization point in its underlying Labeled Transition System (LTS). No manual feature engineering is required.
>
> The confusion may arise from our 'BP-strategies' (Section 3.3), which are indeed manually designed. However, we introduced these explicitly as a baseline to demonstrate that automatic $O_{BP}$  representations outperform hand-crafted features. The core contribution is the automatic extraction; BP-strategies simply validate its superiority over manual alternatives.
>
>
>
> ## W3
>
> Regarding reward modeling and stochasticity (W3): The BP model defines only the environment dynamics (transition function), not the reward. The reward function remains external and follows standard RL conventions - in our experiments, we use the sparse MiniGrid reward detailed in Section 2.2 ($R=1-0.9 \times stepCount / maxSteps$ at goal, 0 otherwise). The 'goal' b-thread (Listing 1) simply signals the termination condition ($terminated=True$), not the reward value.
>
> For stochasticity: BP naturally handles nondeterministic event selection through its arbitrator - when multiple events are requested and not blocked, one is chosen (deterministically in our implementation, though stochastic selection is supported). Environment stochasticity (e.g., randomized layouts) is handled identically to standard MiniGrid, as BP models the logical constraints, not the random initialization.
>
>
> ## W4, Q1, Q4
> Concerns about MiniGrid (W4) and the extensibility to stochastic, partially observable, and continuous domains (Q1, Q4) have been addressed in our general response.

---

### Author Response · Authors · 2025-11-20
**Answering Shared Concerns Of The Reviewers**

## On Full Observability in MiniGrid
A concern was raised that our BPRL-derived observation ($o_{BP}$) provides an unfair advantage. We must clarify that this is a misunderstanding of our baseline. As defined in our Baselines (Section 4.1), the 'Original' observation is *not* the standard partially-observable agent view. Instead, to provide a much stronger baseline, we use the **fully observable** $H \times W \times 3$ grid, which encodes the entire environment's contents, not as RGB but as a more high-level representation (object type, color, and state).

Thus, both our BPRL agent and our baseline agent receive the *exact same* global information. The superior performance of BPRL (seen in Figure 3 and Tables 1 and 2) is therefore not due to receiving more information, but from receiving a superior **representation** of that same information. using the baseline, the agent must learn symbolic concepts from a raw grid tensor (not an RGB-like representation, but a high-level one), whereas with our representation, the agent receives the compact, symbolic state ($o_{BP}$) derived automatically from the environment's formal specification (Section 3.2).

## Partial observability and continuous domains

Reviewers raised valid concerns about extensibility to partially observable (PO) and continuous domains, which we explicitly identify as key directions for future work (Section 6).

For **visual-based PO domains** (e.g., using raw pixels), our "multi-modal state representation'' (Line 53) and **dual-branch network architecture** (Section 4.2, Appendix B) are designed to fuse raw visual input (via a CNN) with our symbolic $o_{BP}$ vector (via an FCN). Our BP observation, along with original experiments, validates this multimodal concept. Handling partial observability at the **behavioral level** is a more complex, non-straightforward research challenge, as it requires the BP model to reflect limited information, and is a key part of our future work.

Regarding **continuous control**, we must distinguish our *experiments* (MiniGrid) from the *formalism*. The BP formalism is mature and well-established in continuous domains, with published work demonstrating its use in controlling physical robotic platforms and its integration with SMT solvers (like Z3) to manage real-valued variables and complex numerical constraints (relevant papers are in our related work section 5.2). Extending our automatic state derivation method to these domains is a primary goal of our future work (Section 6).

## On Scalability via Non-Intrusive Extensibility
A key feature of the BP formalism that enables scalability is "non-intrusive extensibility''. This is a dominant feature of BP that allows system specifications to scale. We provide a concrete example of this in our paper (Section~3.1 and Listings 1--2). We first build the MiniGrid-Empty environment using a modular set of b-threads (wall, goal, etc.). To create the far more complex MiniGrid-DoorKey environment, we do *not* refactor or modify the existing code; we **incrementally add** new, independent b-threads (e.g., *door_unlock_with_key* b-thread). This ability to add complex behaviors without altering existing modules (e.g non-intrusively) is the core of BP's scalability. This translates directly to a scalable representation: when a new b-thread is added to the environment model, our BPRL framework automatically expands the state vector to include this new behavior.

---

### Author Response · Authors · 2025-11-26
**Summary of Revision**

In response to the reviewers' comments, we have incorporated related work from relevant fields to better situate our contribution within the research landscape. Furthermore, we have added **two additional sections** to the appendix: (1) the BP formalism, including a practical example , and (2) supplementary information regarding the integration and implementation of BP-strategies, provided with code examples.

---

### Meta-Review · Area_Chair_kEhJ · 2026-01-07

**Summary:**

This paper proposes using Behavioral Programming (BP), a modeling formalism, to specify environments; having done so, the authors propose a mechanism for automatically constructing state representations. Through experiments on a few MiniGrid domains, the authors demonstrate that the BP-derived state representations help sample efficiency and downstream performance.

Overall I, and most of the reviewers, agree that the idea of using BP for environment specification, and by extension for state representations, is quite interesting and novel. However, the entire paper's argument rests on the assumption that a BP-environment specification is provided, and this needs to be done manually. MiniGrid lends itself rather nicely for this (and the authors provided some clarity on how to do so during the  rebuttal), but it's not clear how easy/natural this would be for other environments. As such, I do not believe the paper is ready for publication yet.

I would recommend the authors first _convincingly_ argue for the practicality of using BP as a general-purpose formalism for environment specification, and _then_ show that this leads to a natural notion of state representation.

**Reviewer Concerns:**

I highlight what I found to be the most important concerns raised by reviewers:

## PEux
- W1 (paper does not adequately justify why BP is a particularly suitable modeling formalism for RL). The reviewer was mostly asking for a comparison with other formalisms such as hierarchical RL, options, etc. The authors rightly pointed out that these are _policy abstractions_, whereas BP is meant for environment specification and as such, is orthogonal. With regards to expressiveness, the authors use their MiniGrid examples to highlight their expressivity. Unfortunately they do not discuss how applicable it would be to other envs (since the reviewer did not ask for it explicitly).
- W2/W3 (technical definitions of BP and how stochasticity/rewards are handled). The authors do provide an answer for this, but it highlights how the paper is somewhat lacking in terms of justifying the use of BP in the first place.
- W4 (only experiments on MiniGrid). The authors provided a general response to this, but it is mostly about how BP _could_ be used for other environment types, without providing explicit examples. As mentioned above, I believe this is a key requirement in order to properly justify the use of BP in the first place.
- W5 (why BP-based representations help). The authors provide a justification for this, but it is somewhat high-level and "hand-wavy", and relying on the MiniGrid examples too much. I don't feel this was properly addressed, as this justification should really be general-purpose.

## RXZR
- W1 (may be difficult to extend [BP] to more complex or unknown environments). The authors respond that writing b-threads is manual, but "this effort is for environment modeling (Section 3.1), a prerequisite step in any RL setup". This is not a great response, as MiniGrid (and most benchmarks used) already have an environment specification. The authors should be providing examples of how to use BP in other environment types. The reviewer also raised a point about "unknown environments", which is a good one (i.e. most real-world systems would be mostly unknown). The authors do not really respond to this concern.
- W2 (this paper do not compare with other popular learning-based state representations such as bisimulation metric and BP-based methods mentioned in the related work.). The authors respond that "methods like bisimulation are complementary". While true to some extent, this is a valid reviewer concern. Given that the authors have both the original and BP-derived environment specifications for MiniGrid, they could compare BP-derived representations (with the BP-env specification) with bisimulation (or other) learned representations (with the original env specification); this would enable a comparison of learning efficiency, which could help strengthen the claims of BC as a useful formalism for environment specification.
- Q2 (Is there a more automated and generalizable way to write BP code?). The authors provide some ideas for how to do this, but it would have been better to provide concrete examples of this being done, so as to better justify the use of BP.

## Jpnj
- Main weakness (BP-based observations contain privileged information that the agent ideally should not have access to). This is a valid concern, especially when arguing for BPRL's efficiency. The authors respond that "our 'Original' observation is not the standard partially-observable agent view. Instead, to provide a much stronger baseline, we use the fully observable HxWx3 grid, which encodes the entire environment's contents. Thus, both our BPRL agent and our baseline agent receive the exact same global information.". This is not entirely true, as BP highlights the important bits (i.e. whether the agent has a key), whereas the full raw 'original' observation has everything bundled in, so it is not really a fair comparison. As such, the resulting BP-representation _does_ have a bit of an unfair advantage.
- C1 (Once the full environment dynamics are programmed, deriving a proper state representation is relatively straightforward). The authors respond that this not quite true, and their BP-based representations are derived automatically. I believe this adequately addresses the reviewer concern.
- Q4 (comparison against other Markovian representations). The authors respond that they compared against the 'Original' state inputs, as well as manually-specified BP-based representations. While this partially addresses the reviewer concern, I feel more could have been done to better justify BPRL, as discussed in the point about bisimulation raised by RXZR.

## V7yu
- W1 (the paper meanders between two different topics: (1) getting state observations from BP environments and (2) specifying advice using BP-strategies). The authors correctly respond that "(1) is our core contribution, and (2) is our primary ablation baseline", which is correct.
- W2 (Comparing these agents means something very different, since the BP-strategies agent contains additional heuristics. It’s
genuinely hard to make heads or tails of what is useful here to compare, it’s currently comparing apples to apples to oranges to guavas). The authors respond that "the comparison in Table 1 is not "apples to oranges". It is a direct, empirical comparison between our automatic method (BP-observation + Original) and a strong manual alternative (Original + strategies)". While this is true, it relates to previous points made by other reviewers regarding comparisons to other representation learning methods. As such, I feel this strong concern is only partially addressed.
- W3 (Hyperparameter choices). The reviewer was concerned that hparams obfuscate algorithmic differences, but the authors respond that "tuning was conducted extensively but only on the 'Original' baseline", which is good.
- W4 (Scaling of BP). The authors point to their general response, which was already discussed above (and, in my opinion, does not sufficiently address the concern).

**Reviewer Scores:**

- **PEux:** Currently a 4, unlikely to increase given the details above.
- **RXZR:** Currently a 4, unlikely to increase given the details above.
- **Jpnj:** Currently a 4, unlikely to increase given the details above.
- **V7yu:** Currently a 2, unlikely to increase given the details above.

---

### Decision · Program_Chairs · 2026-01-26

Reject